# DIsoN: Decentralized Isolation Networks for Out-of-Distribution Detection in Medical Imaging

**Felix Wagner[1]**     **Pramit Saha[*1]**     **Harry Anthony[*1]**
**J. Alison Noble[1]**     **Konstantinos Kamnitsas[1]**
[1]Department of Engineering Science, University of Oxford
`felix.wagner@eng.ox.ac.uk`

## Abstract

Safe deployment of machine learning (ML) models in safety-critical domains such as medical imaging requires detecting inputs with characteristics not seen during training, known as out-of-distribution (OOD) detection, to prevent unreliable predictions. Effective OOD detection after deployment could benefit from access to the training data, enabling direct comparison between test samples and the training data distribution to identify differences. State-of-the-art OOD detection methods, however, either discard the training data after deployment or assume that test samples and training data are centrally stored together, an assumption that rarely holds in real-world settings. This is because shipping the training data with the deployed model is usually impossible due to the size of training databases, as well as proprietary or privacy constraints. We introduce the **Isolation Network**, an OOD detection framework that quantifies the difficulty of separating a target test sample from the training data by solving a binary classification task. We then propose **Decentralized Isolation Networks (DIsoN)**, which enables the comparison of training and test data when data-sharing is impossible, by exchanging only model parameters between the remote computational nodes of training and deployment. We further extend DIsoN with class-conditioning, comparing a target sample solely with training data of its predicted class. We evaluate DIsoN on four medical imaging datasets (dermatology, chest X-ray, breast ultrasound, histopathology) across 12 OOD detection tasks. DIsoN performs favorably against existing methods while respecting data-privacy. This decentralized OOD detection framework opens the way for a new type of service that ML developers could provide along with their models: providing remote, secure utilization of their training data for OOD detection services. Code available at: `https://github.com/FelixWag/DIsoN`

## 1 Introduction

Consider the standard setting where an organization, such as a software company, develops and trains a Machine Learning (ML) model to perform a task of interest, for example disease classification in medical images, using a training database. The organization then provides the model to a user (e.g., a client) and deploys it, for example in a hospital, to make predictions on new *test* samples. Because of heterogeneity in real-world data or user-error, a deployed model may receive inputs unlike anything seen in the training data. Such inputs are called out-of-distribution (OOD) samples, in contrast to the in-distribution (ID) samples that follow the distribution of the training data. For example, OOD patterns in medical imaging can be artifacts from suboptimal acquisition or unknown diseases. Performance of ML models often degrades unexpectedly on OOD data [43]. Thus, to ensure safe deployment in safety-critical applications such as healthcare, deployment frameworks should

---

[*]Equal second-author contribution.

39th Conference on Neural Information Processing Systems (NeurIPS 2025).

include mechanisms to detect OOD inputs, so that they can be flagged to human users and avoid adverse effects of using potentially wrong model predictions in the downstream workflow.

Multiple OOD detection methods have been previously developed, which can be broadly categorized into *post-hoc* and *training-time regularization* methods [42]. Post-hoc methods assume that a "primary" model has been trained and deployed to perform a task of interest, such as disease classification, and they aim to perform OOD detection without alterations to this primary model [13, 22, 23, 27, 15]. On the other hand, training-time regularization methods alter the embedding space of the primary model during its training, to allow improved OOD detection during inference [30, 29, 5, 14, 36, 28].

Few OOD detection methods leverage direct comparison of test-samples against the training data to identify differences between them for OOD detection, such as KNN-based [34] or density-based methods [6]. These require, however, the training data to be available at the site of deployment, for direct comparison with test-samples. Therefore they are impractical in many applications where sharing and storing the training samples or their embeddings at each site of deployment is impossible due to privacy, legal, or proprietary constraints. Because of this, most other methods do not compare the test-samples against the training data [13, 21] or make indirect comparisons using auxiliary representations of training data, such as via summary statistics [22], prototypes [29, 30] or synthetic samples [7]. Such derived representations, however, do not faithfully capture all intricacies of the original training data. Hypothesizing that direct comparison with the original training data would be useful to infer whether a test sample is OOD, this paper addresses the following question:

> *Can we design an OOD detection algorithm that compares test samples against the original training data, without requiring transfer of training data to the point of deployment?*

This paper describes a novel OOD detection framework (Fig. 1) with the following key aspects: (1) We introduce **Isolation Networks** for OOD detection. To infer whether a new test sample is OOD, a neural network is trained to learn a binary classification boundary that separates (isolates) a test sample from the original ID training samples. The network's convergence rate is then used as a measure of whether the target sample is OOD or similar to the training data. The intuition is that test samples with OOD patterns will be easier to separate from the training samples than ID test samples without OOD patterns. Isolation Networks draw inspiration from Isolation Forests [25], which train decision trees to isolate each training sample. To infer whether a test-sample is OOD, the number of split nodes needed to isolate it is used (c.f. Sec. 2). Isolation Networks instead train a different network for each test sample, to separate it from the training data, and measure convergence as OOD score. (2) We then introduce **Decentralized Isolation Networks (DIsoN)**, a decentralized training framework that enables training Isolation Networks in the practical setting when the ID training data are held on a different computational node than the node of deployment, where the test samples are processed. (3) We extend DIsoN to class-conditional training, where the class of a test sample is first predicted and then used to compare it only against samples in the ID training database of the same class. This reduces variability within the distribution of training samples that the model uses for comparison, making it harder to isolate ID samples that closely resemble their class than OOD samples, improving OOD detection.

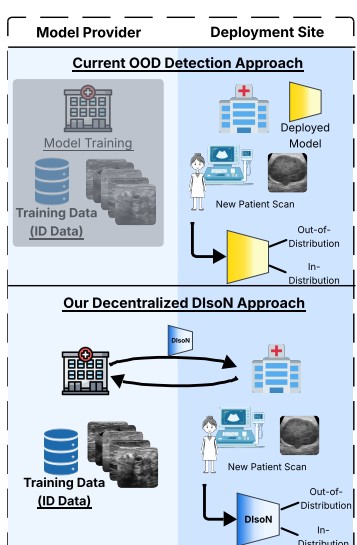

Figure 1: **(Top)** Most OOD detection methods do not use training data after deployment because it cannot be shipped with the model. **(Bottom)** DIsoN enables decentralised comparison of test samples with the training data via model parameter exchange.

We evaluate the capabilities of DIsoN and state-of-the-art OOD detection methods in identifying OOD patterns that occur in real-world applications. For this, we experiment on 4 medical imaging datasets, including dermatology, chest X-rays, breast ultrasound and histopathology, where we

evaluate 12 OOD detection tasks. Results show DIsoN outperforms current state-of-the-art methods, demonstrating the effectiveness of remotely obtaining information from the training data during inference to identify unexpected patterns in test samples, while adhering to data-sharing constraints.

## 2 Related Work

**OOD Detection methods.** *Post-hoc methods* for OOD detection assume a "primary" model pre-trained for a task of interest, such as disease classification, and aim to perform OOD detection without modifying it. A common baseline is using Maximum Softmax Probability (MSP) [13] as a measure of whether a test sample is OOD. Another common approach is to represent the training data via summary statistics, such as using class-conditional Gaussians in the model's latent space, and use Mahalanobis distance to detect OOD samples [13]. In a similar fashion, other post-hoc approaches use the logit space (energy score [27]), gradient space, (GradNorm [15]) and feature space (fDBD [26]), or combine multiple representations (ViM [37]) to derive a measure for OOD detection. Another group of methods use *training-time regularization*. Regularization modifies the embedding space of the primary model during training to bolster OOD detection performance at inference [14, 5, 39, 9]. A representative example of the state-of-the-art is CIDER [30], which uses a loss function to optimize the model's feature space for intra-class compactness and inter-class dispersion. Recently, PALM [29] this further by modeling each class with multiple prototypes.

Most OOD detection methods, such as the above, do not directly utilize the training data after the model's deployment. Thus they do not directly compare them with the test samples processed after deployment, which could facilitate detecting patterns that differentiate them. There are few notable exceptions, such methods based on KNN [34] and Local Outlier Factor [6]. These compare an input's embedding to the training data embeddings. They require, however, shipping and storing the training data or their feature vectors to each deployment site, which in many practical applications can be infeasible due to privacy constraints or the size of training databases.

Inspiration for this work was drawn from the seminal work on Isolation Forests (iForest [25]), which has received multiple extensions such as Deep Isolation Forests [40]. They train decision trees using the original ID training data or their deep embeddings respectively, to learn partitions that isolate each sample. To infer whether a test-sample is OOD, they count the number of split nodes applied by the trained trees until it is isolated. OOD samples should need less splits than ID nodes. The algorithm introduced herein trains a neural network classifier to isolate a single test-sample from the ID training data, focusing on extracting patterns that distinguish the specific sample. We demonstrate the use of convergence rate as scoring function to infer whether a test-sample is OOD. We also demonstrate the use of decentralised training to enable such an isolation algorithm without data-sharing.

**Federated Learning and OOD Detection.** Training of DIsoN uses decentralised optimization similar to Federated Learning (FL). Using FL for OOD detection is a recent area. The few related works, such as [44, 24], studied settings significantly different from ours: they assume the training data is decentralized across multiple computational nodes, where each node's data follow a different distribution. Using FL, they train a model that measures distribution-shifts between nodes. After training, the model is applied on test-samples for OOD detection. Our algorithm does not require training data from multiple distributions or multiple nodes. It assumes one node holds all ID training data and a second node holds a single test-sample, which we infer whether it is OOD via decentralised training of a binary classifier. Thus the motivation, use-case and technical challenges are distinct. Moreover, DIsoN may resemble FL with a few-shot client [33, 38]. Few-shot methods are designed to regularize against overfitting the limited data of few-shot client(s), to train a model for a predictive task (e.g. disease diagnosis) that generalizes to new samples. DIsoN does not train a model for generalization. It optimizes for distinguishing data of the source node from a target sample, using convergence speed to infer if the latter is OOD. Hence, few-shot methods are not directly applicable.

## 3 Method

**Preliminary: Out-of-Distribution Detection.** Let $\mathcal{X}$ be the input data space and $\mathcal{Y} = \{1, 2, \ldots, C\}$ the label space for a classification task with $C$ classes. We denote the *source training dataset* as $\mathcal{D}_s = \{(\mathbf{x}_i, y_i)\}_{i=1}^{N}$, assumed *i.i.d.* from the joint distribution $\mathbb{P}_{\mathcal{X}\mathcal{Y}}$. The marginal distribution over the input data space $\mathcal{X}$ is denoted as $\mathbb{P}_{\mathcal{X}}^{in}$. The goal of OOD detection is to determine, during

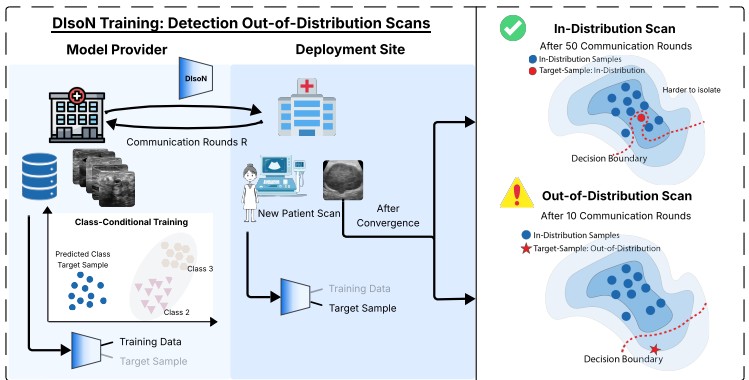

Figure 2: **Overview of DIsoN**: When a new scan is obtained at the deployment site, DIsoN is trained using parameter updates from both the deployment site and the model provider, who holds the ID training data, to isolate the target sample from the training data. The deployment site trains on the single target scan, while the model provider trains on the source data (optionally class-conditioned). Only model parameters are exchanged. **(Right)** After convergence, the scan is classified as OOD if it is isolated in few rounds, and as ID otherwise.

inference, whether a _target sample_ $\mathbf{x}_t \in \mathcal{X}$ originates from the source distribution $\mathbb{P}^{in}_{\mathcal{X}}$, or an unknown OOD distribution $\mathbb{P}^{out}_{\mathcal{X}}$. This can be achieved by defining an OOD scoring function $S : \mathcal{X} \to \mathbb{R}$, that assigns high scores to ID samples and low scores to OOD samples. We label a sample $\mathbf{x}$ as OOD when the scoring function $S(\mathbf{x}) < s^\star$, and ID if $S(\mathbf{x}) \geq s^\star$, with a chosen threshold $s^*$.

### 3.1 Isolation Networks

The core idea of our approach is that the difficulty of training a binary classifier to separate a single target sample $\mathbf{x}_t$ from the source data $\mathcal{D}_s$ gives an indication whether $\mathbf{x}_t$ is ID or OOD. Intuitively, a target sample that is ID will share characteristics with samples in $\mathcal{D}_s$. As a result, it is _harder_ for a classifier to isolate (separate) the target sample and will require more update steps. On the other hand, an OOD target sample has patterns that differ from $\mathcal{D}_s$, making it _easier_ to isolate and requiring fewer update steps during training. Fig. 2 gives a visual intuition of the idea.

Formally, we consider a neural network consisting of a feature extractor $f : \mathcal{X} \to \mathcal{Z}$, parameterized by $\theta^f$, and a binary classification head $h : \mathcal{Z} \to [0,1]$, with parameters $\theta^h$. The full network is parameterized by $\theta = (\theta^f, \theta^h)$. We use $m \in \{0, 1\}$ for isolation labels and reserve $y$ for class labels. In an _idealized_ centralized setting with full access to $\mathcal{D}_s$ and the target sample $\mathbf{x}_t$, we can train the network $h \circ f$ for the following binary classification problem: Assign label $m = 0$ to all source samples $\mathbf{x}_s \in \mathcal{D}_s$ and label $m = 1$ to the target sample $\mathbf{x}_t$. Typically, $\mathcal{D}_s$ has a high number of samples compared to the single target sample $\mathbf{x}_t$, $|\mathcal{D}_s| \gg 1$. Therefore, directly training on this highly imbalanced setup is suboptimal, as it can lead to a trivial classifier that always predicts the majority class ($m = 0$), ignoring the minority class ($m = 1$). To ensure that the target sample has an impact during training, we _over-sample_ $\mathbf{x}_t$ within each mini-batch. Specifically, we construct mini-batches as $B = B_s \cup \{\mathbf{x}_t^{(1)}, \ldots, \mathbf{x}_t^{(N)}\}$, where $B_s$ is a set of source samples drawn from $\mathcal{D}_s$, and $\{\mathbf{x}_t^{(1)}, \ldots, \mathbf{x}_t^{(N)}\}$ are $N$ replicated copies of the same target sample $\mathbf{x}_t$. The empirical loss for this optimization objective is:

$$\mathcal{L}_c(\theta; B_s, \mathbf{x}_t, N) = \frac{1}{|B_s| + N} \left( \sum_{\mathbf{x}_s \in B_s} L(\theta; \mathbf{x}_s, m = 0) + N \cdot L(\theta; \mathbf{x}_t, m = 1) \right), \quad (1)$$

where $L(\theta; x, m)$ denotes the binary cross-entropy loss. In Sec. 3.2 we compare the gradient of the centralized version $g_c(\theta; B_s, \mathbf{x}_t, N) = \nabla_\theta \mathcal{L}_c(\theta; B_s, \mathbf{x}_t, N)$ to our proposed decentralized version.

**Convergence Time as OOD Score.** To quantify the "isolation difficulty", we track the number of training steps required by the binary classifier to separate $\mathbf{x}_t$ from $\mathcal{D}_s$. Let $\theta^{(k)}$ be the model parameters after $k$ optimization steps with a stochastic gradient-based optimizer (e.g. SGD or Adam [20]) using the gradients from the loss of Eq.1. Formally, we define our convergence time $K$ as:

**Definition 3.1** (Convergence Time $K$). *Let $p^{(k)}(x) = h(f(x; \theta^{(k)}))$ be the sigmoid-score after $k$ updates. Fix a window $E_{\text{stab}} \in \mathbb{N}$ and accuracy threshold $\tau \in (0, 1]$. The convergence time $K$ is the smallest $k \geq E_{\text{stab}}$ such that*

$$\min_{j \in [k - E_{\text{stab}} + 1, \, k]} p^{(j)}(x_t) > 0.5 \quad and \quad \frac{1}{|\mathcal{D}_s|} \sum_{x_s \in \mathcal{D}_s} \mathbb{I}\{p^{(k)}(x_s) \leq 0.5\} \geq \tau.$$

Intuitively, the conditions ensure that the classifier correctly classifies $x_t$ for the last $E_{\text{stab}}$ consecutive updates (first condition), while achieving an accuracy of at least $\tau$ on the source data (second condition). We set $E_{\text{stab}} = 5$ and $\tau = 0.85$. Further details on these parameter choices are provided in the Appendix D. We define our OOD scoring function as $S(x_t) = K$, following the convention where ID samples have higher scores. We emphasize that this setup corresponds to the centralized version of the algorithm, where both $D_s$ and $x_t$ are accessible on the same node. In Sec. 3.2, we extend it to the decentralized case, where source data and the target sample reside on separate nodes.

## 3.2 Decentralized Isolation Networks (DIsoN)

**Problem Setting: Decentralized OOD Detection.** The Isolation Network described above assumes direct access to both the source data $\mathcal{D}_s$ and the target sample $x_t$. However, as described in Sec.1, this assumption does not hold in many real-world settings. We therefore formalize a decentralized setting involving two sites: the *Source Node* (SN) and the *Target Node* (TN). SN represents a model provider that holds the source training dataset $\mathcal{D}_s$, on which it has pre-trained a primary model $M_{pre}$ for the main task of interest, such as a $C$-class classifier of disease. $M_{pre}$ consists of a feature extractor (parameterized by $\theta^{pre}$) and a $C$-class classification head. TN holds target samples $x_t$ and represents the site where $M_{pre}$ is deployed to process $x_t$. Therefore, the aim of the OOD detection algorithm is to infer whether a given $x_t$ on TN is ID or OOD, to support safe operation of $M_{pre}$. To this end, TN can exchange model parameters with SN, but not raw data or their embeddings due to privacy and regulatory constraints.

**DIsoN Method Overview.** To approximate the training dynamics of an Isolation Network (Sec. 3.1) without data sharing and without requiring centralized access to both the source data and target samples, we propose DIsoN. Fig.2 shows an overview of DIsoN. Inspired by the Federated Learning framework, our algorithm trains parameters of an Isolation Network over multiple communications rounds between SN and TN. We keep track of a *global* set of Isolation Network parameters. At the start of each round $r$, they are transmitted to SN and TN. In each round $r$, both SN and TN update them locally by performing $E \geq 1$ local optimization steps. At the end of a round, the global set of parameters is updated via a weighted aggregation of the local updates. In more detail:

**1) Initialization.** Let $\theta^{(r)}$ be the global model parameters of the Isolation Network at the start of round $r$. The feature extractor $\theta^f$ of the initial global model $\theta^{(0)} = (\theta^f, \theta^h)$ is initialized with the pre-trained parameters $\theta^{pre}$ of primary model $M_{pre}$, while the binary classification head is initialized randomly. This initialization is done on SN, after which $\theta^{(0)}$ is transmitted to TN.

**2) Local Updates.** Each site performs local training for $E$ steps using its own data: The **Source Node** initializes its local model $\theta_S^{(r,0)} = \theta^{(r)}$ and performs $E$ optimization steps on mini-batches $B_s \sim \mathcal{D}_s$, minimizing $L(\theta; x_s, 0)$, resulting in $\theta_S^{(r,E)}$. Similarly, the **Target Node** initializes its local model $\theta_T^{(r,0)} = \theta^{(r)}$ and performs $E$ optimization steps on the single target sample $x_t$, minimizing $L(\theta; x_t, 1)$, resulting in $\theta_T^{(r,E)}$.

**3) Model Aggregation.** After local updates, TN sends $\theta_T^{(r,E)}$ back to SN. The models are then aggregated into an updated global model using weighted averaging:

$$\theta^{(r+1)} = \alpha \cdot \theta_S^{(r,E)} + \beta \cdot \theta_T^{(r,E)}, \tag{2}$$

where the aggregation weights are $\beta = 1 - \alpha$. The updated global model $\theta^{(r+1)}$ is then sent to TN to start the next local training round.

Before each local training round starts, we evaluate the current global model $\theta^{(r)}$ using the convergence criteria in Def. 3.1 (TN evaluates on $x_t$, SN on $\mathcal{D}_s$). If converged, we record $R = r$ and terminate. The OOD score for the target sample $x_t$ in the decentralized setting is based on the

number of communication rounds $R$ required to converge, analogous to the number of steps $K$ in the centralized case: $S_{\text{DIsoN}}(\mathbf{x}_t) = R$. We can draw a connection between DIsoN and our centralized version. We show that under specific conditions, the decentralized and the centralized versions result in equivalent model parameters:

**Proposition 3.1** (DIsoN and Centralized Isolation Network Equivalence for $E = 1$). *Let $\theta_{cent}$ be the model parameters from our centralized algorithm and $\theta_{dec}$ be the parameters from DIsoN. Let each site perform one local SGD step ($E = 1$) with learning rate $\eta$, and aggregate with $\alpha = \frac{|B_s|}{|B_s|+N}$, $\beta = \frac{N}{|B_s|+N}$. Then the decentralized update equals the centralized one:*

$$\theta_{\text{dec}}^{(r+1)} = \theta^{(r)} - \eta\big(\alpha\, g_S(\theta^{(r)}) + \beta\, g_T(\theta^{(r)})\big) = \theta^{(r)} - \eta\, g_c(\theta^{(r)}) = \theta_{\text{cent}}^{(r+1)},$$

*where $g_S(\theta) = \frac{1}{|B_s|}\sum_{\mathbf{x}_s \in B_s} \nabla_\theta L(\theta; \mathbf{x}_s, 0)$, $g_T(\theta) = \nabla_\theta L(\theta; \mathbf{x}_t, 1)$ and $N$ is the number of times $\mathbf{x}_t$ is oversampled in the centralized version.*

*Proof sketch.* Insert $\theta_S^{(r,1)} = \theta^{(r)} - \eta g_S(\theta^{(r)})$, $\theta_T^{(r,1)} = \theta^{(r)} - \eta g_T(\theta^{(r)})$ into the weighted average; factor out $\theta^{(r)}$, use $\beta = 1 - \alpha$. Full derivation can be found in the Appendix. $\square$

Prop. 3.1 shows that our decentralized training exactly replicates the centralized version when $E = 1$ and aggregation weights are chosen accordingly. This theoretical equivalence motivated our design of DIsoN: it preserves the core idea of the Isolation Network, while meeting our decentralized data sharing constraints. In practice, we allow $E > 1$ to reduce communication overhead, which causes the decentralized updates to deviate from the centralized version. However, as we show in Sec. 4, DIsoN achieves promising results even with the approximation. We found DIsoN to be robust across a broad range of $\alpha$ values, as shown in an ablation study in Sec. 4.2.

**Practical Techniques: Augmentation and Normalization.** One challenge in DIsoN is to avoid rapid memorization and overfitting of superficial features in the target sample $\mathbf{x}_t$, making the isolation task trivial regardless whether the sample is OOD. To prevent this, we apply stochastic data augmentations (e.g. random crops, horizontal flips), which regularize the model to learn invariant features, so that the separation is based on semantic characteristics. Furthermore, in DIsoN, we use Instance Normalization (IN) [35] instead of the widely used Batch Normalization (BN) [16] layers for feature normalization. BN relies on batch-level statistics, which are not suitable for our single-sample TN. IN instead normalizes each feature map per sample and therefore fits better for our decentralized, single target sample scenario.

### 3.3 Class-Conditional Decentralized Isolation Networks (CC-DIsoN)

To further improve DIsoN, we introduce its class-conditional variant, **CC-DIsoN**. The intuition is that ID samples should be especially hard to isolate from source samples of the same class, as they are likely to share similar visual features compared to samples from other classes. To use this idea, we modify the source data sampling strategy during local training at SN. After the initialization phase, TN uses the pre-trained model $M_{pre}$ to predict the class of the target sample: $\hat{y} = \arg\max_c [M_{pre}(\mathbf{x}_t)]_c$. Afterwards, TN sends the predicted label $\hat{y}$ to SN. During local training at SN, mini-batches are now sampled *only* from source data from class $\hat{y}$: $B_s \subset \{(\mathbf{x}_s, y_s) \in \mathcal{D}_s \mid y_s = \hat{y}\}$. The other steps of our method remain the same. Our empirical results in Sec. 4.2 confirm that this improves OOD detection.

## 4 Experiments & Results

**Datasets.** We evaluate DIsoN on four publicly available medical imaging benchmark datasets covering dermatology, breast ultrasound, chest X-ray, and histopathology. All datasets consist of real, clinically acquired images and no synthetic data is used. The first three benchmarks use images with naturally occurring non-diagnostic artifacts as OOD samples (e.g., rulers, pacemakers, annotations), while histopathology focuses on semantic and covariate shifts across domains. Example images are shown in Fig.3.

**Dermatology & Breast Ultrasound:** We adopt the benchmark setup from [3], using images without artifacts as the training and ID test data, and images with artifacts (rulers and annotations) as OOD samples. For breast ultrasound (BreastMNIST [41]), the artefacts are embedded text annotations, and

for dermatology (D7P [19]) the artefacts are black overlaid rulers. For breast ultrasound, the primary model $M_{pre}$ is trained for 3-class classification (normal/benign/malignant). The 228 annotated scans with artifacts are used as OOD test samples, while the remaining artifact-free scans are split 90/10 into training and ID test sets. For dermatology, the primary model $M_{pre}$ is trained for binary classification (nevus/non-nevus). The annotated 251 images with rulers are used as OOD samples and the remaining 1403 are split 90/10. Images are resized to $224 \times 224$.

**Chest X-Ray:** Following the benchmark from [2], we use frontal-view X-ray scans (from CheXpert [17]) containing no-support devices as the training and ID test data, and scans containing pacemakers as OOD samples. The primary model $M_{pre}$ was trained for the binary classification of cardiomegaly. We use the 23,345 annotated scans without any support devices as our training data, and randomly hold out 1000 ID samples for testing. The OOD test set includes 1000 randomly sampled scans with pacemakers. All images are resized to $224 \times 224$.

**Histopathology:** We use the MIDOG benchmark from OpenMIBOOD [11]. The MIDOG dataset [4] consists of $50 \times 50$ image patches extracted from Hematoxylin & Eosin-stained histological whole-slide images, grouped into different "domain" data sets corresponding to different imaging hardware, staining protocols, or cancer types. The primary model is trained for 3-class classification (mitotic/imposter/other) on domain 1a. Following [4] we evaluate on two settings: (i) **Near-OOD**: Domains 2-7 are treated as separate OOD detection task with only moderate domain shifts. (ii) **Far-OOD**: Using CCAgT [1] and FNAC 2019 [32] dataset as the OOD task, which differ significantly due to being completely different medical applications. We use the 251 test ID samples from domain 1a and randomly sample 500 OOD samples from each of the near- and far-OOD domains.

**Training Details.** We use a ResNet18 [12] with Instance Normalization (as per Sec.3.2) pre-trained on the dataset-specific task as initialization for DIsoN. DIsoN is trained with Adam (lr=0.001 for dermatology and ultrasound; 0.003 for X-ray). For histopathology SGD with momentum (lr=0.01, momentum=0.9) is used (since [11] suggests pretraining with SGD). Local iterations per communication round are chosen to approximately match one epoch on the training data. We use standard augmentations (e.g. random cropping, rotation, color-jitter) and the aggregation weight is fixed to $\alpha = 0.8$, since it performs consistently well across all our experiments (see Sec. 4.2 for effect of $\alpha$). Experiments were run on an NVIDIA RTX

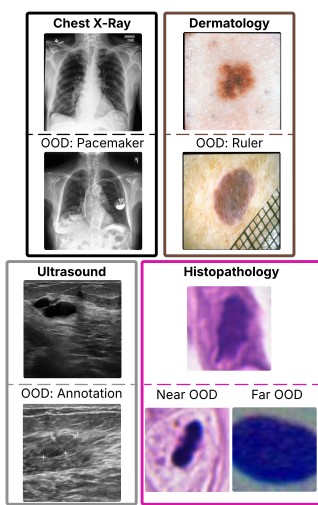

Figure 3: **Examples of data. X-Ray:** ID X-ray vs. OOD scan with pacemaker. **Dermatology:** ID lesion vs. OOD image with ruler. **Ultrasound:** ID artifact-free ultrasound vs. OOD scan containing annotations. **Histopathology:** ID mitotic-cell patch vs. near-OOD patch with different cancer type and far-OOD patch with different staining.

A5000. More training details and hyperparameters are provided in the Appendix. OOD detection performance is evaluated with two metrics: (i) area under the receiver operating characteristics curve (AUROC, higher is better) (ii) the false positive rate at 95% true positive rate (FPR95, lower is better). All results in Tables 1 and 2 are averaged over three runs with different seeds, and we report mean and standard deviation. Standard deviations for Tab.2 are in Appendix I due to space limitations.

**Baselines.** We compare our method against state-of-the-art OOD detection methods from the two main categories: *post-hoc* and *training-time regularization* methods. The post-hoc methods include: MSP [13], MDS [22], fDBD [26], ViM [37], Deep iForest [40]. For *training-time regularization methods*, we evaluate the recent methods CIDER [30] and PALM [29] (both using contrastive learning). We use MDS for OOD scoring on their learned feature representations. Several baselines already use a form of class-conditioning based on the model's predicted class: MDS, PALM, and CIDER via Mahalanobis distances to class clusters; ViM uses the maximum logit of the predicted class; MSP via softmax probability; and fDBD via distance to the decision boundary (all implicitly or explicitly conditioned on the predicted class). Only iForest does not. CC-DIsoN similarly relies solely on the predicted class from $M_{pre}$ for class-conditional sampling and uses no ground-truth labels, providing no additional information beyond these methods.

Table 1: OOD detection performance evaluated on three OOD datasets: Chest X-Ray, Dermatology, and Breast Ultrasound and reported as mean $\pm$ standard deviation over three random seeds. $\downarrow$ means smaller is better and $\uparrow$ means larger is better. **Bold** numbers highlight the best results, second best results are underlined.

| Method | Chest X-Ray | | Dermatology | | Breast Ultrasound | | Average | |
|---|---|---|---|---|---|---|---|---|
| | AUROC$\uparrow$ | FPR95$\downarrow$ | AUROC$\uparrow$ | FPR95$\downarrow$ | AUROC$\uparrow$ | FPR95$\downarrow$ | AUROC$\uparrow$ | FPR95$\downarrow$ |
| MSP | $60.44_{\pm4.2}$ | $100.00_{\pm0.0}$ | $65.39_{\pm3.9}$ | $100.00_{\pm0.0}$ | $58.85_{\pm4.5}$ | $100.00_{\pm0.0}$ | $61.56_{\pm2.0}$ | $100.00_{\pm0.0}$ |
| MDS | $53.82_{\pm1.5}$ | $90.47_{\pm1.0}$ | $69.36_{\pm5.5}$ | $76.06_{\pm15.5}$ | $61.02_{\pm1.7}$ | $74.40_{\pm1.0}$ | $61.40_{\pm2.5}$ | $80.31_{\pm5.5}$ |
| fDBD | $68.26_{\pm0.4}$ | $78.07_{\pm5.4}$ | $63.59_{\pm2.5}$ | $88.26_{\pm10.8}$ | $60.73_{\pm2.1}$ | $87.50_{\pm4.7}$ | $64.19_{\pm1.6}$ | $84.61_{\pm5.5}$ |
| ViM | $62.60_{\pm4.8}$ | $85.80_{\pm7.3}$ | $68.39_{\pm2.0}$ | $75.35_{\pm6.1}$ | $59.44_{\pm2.8}$ | $\mathbf{73.21_{\pm3.1}}$ | $63.48_{\pm1.5}$ | $78.12_{\pm4.2}$ |
| iForest | $56.35_{\pm5.6}$ | $88.27_{\pm5.1}$ | $56.68_{\pm6.8}$ | $87.31_{\pm3.7}$ | $47.02_{\pm4.8}$ | $95.82_{\pm2.1}$ | $53.35_{\pm3.3}$ | $90.47_{\pm1.5}$ |
| CIDER | $70.47_{\pm6.3}$ | $79.00_{\pm9.9}$ | $81.98_{\pm2.8}$ | $56.57_{\pm8.2}$ | $58.03_{\pm3.1}$ | $82.73_{\pm5.2}$ | $70.16_{\pm3.4}$ | $72.77_{\pm2.5}$ |
| PALM | $65.41_{\pm6.5}$ | $85.73_{\pm9.2}$ | $77.25_{\pm1.0}$ | $62.44_{\pm0.8}$ | $59.35_{\pm3.1}$ | $75.60_{\pm5.5}$ | $67.34_{\pm1.4}$ | $74.59_{\pm1.9}$ |
| **CC-DIsoN** | $\mathbf{84.94_{\pm0.9}}$ | $\mathbf{61.85_{\pm1.2}}$ | $\mathbf{89.54_{\pm1.5}}$ | $\mathbf{42.49_{\pm4.4}}$ | $\mathbf{65.62_{\pm1.2}}$ | $\mathbf{73.21_{\pm4.7}}$ | $\mathbf{80.00_{\pm0.7}}$ | $\mathbf{59.20_{\pm3.1}}$ |

Table 2: OOD detection performance across nine different OOD detection task of MIDOG, split into near-OOD and far-OOD setting. Each cell shows AUROC$\uparrow$/FPR95$\downarrow$, reported as mean over three random seeds. **Bold** numbers highlight the best results, second best results are underlined.

| Methods | near-OOD | | | | | | | | far-OOD | | |
|---|---|---|---|---|---|---|---|---|---|---|---|
| | 2 | 3 | 4 | 5 | 6a | 6b | 7 | Avg. | CCAgT | FNAC | Avg. |
| MSP | 54.1/94.8 | 47.4/93.4 | 55.7/89.4 | 59.0/93.5 | 54.8/94.7 | 45.6/95.5 | 56.1/92.4 | 53.3/93.4 | 78.4/52.2 | 82.7/63.6 | 80.6/57.9 |
| MDS | 67.4/80.0 | 67.2/79.8 | 65.7/81.9 | 61.2/87.4 | 60.6/87.4 | 55.5/89.5 | 51.0/91.6 | 61.2/85.4 | 87.1/43.6 | 86.6/39.6 | 86.8/41.6 |
| fDBD | 57.9/83.0 | 52.5/81.4 | 57.5/88.7 | 60.2/89.0 | 57.9/85.9 | 51.2/86.2 | 55.2/92.2 | 56.1/86.6 | 74.6/58.3 | 82.1/63.7 | 78.3/61.0 |
| ViM | 68.0/79.8 | 66.7/77.8 | 66.2/77.7 | 63.5/86.9 | 61.9/87.3 | 55.7/91.0 | 54.9/93.1 | 62.4/84.8 | 92.5/29.1 | 92.6/27.0 | 92.6/28.0 |
| iForest | 37.9/98.3 | 38.6/96.9 | 39.7/97.7 | 41.6/97.5 | 39.5/98.0 | 40.4/97.7 | 46.4/95.7 | 40.6/97.4 | 27.7/99.2 | 32.3/96.8 | 30.0/98.0 |
| CIDER | 71.4/84.6 | 65.6/89.0 | 55.5/91.5 | 64.2/89.8 | 57.4/92.2 | 47.7/96.5 | 64.1/88.2 | 60.9/90.2 | 82.5/77.2 | 95.4/18.3 | 89.0/47.8 |
| PALM | 73.5/78.1 | 59.2/90.3 | 66.3/77.7 | 64.3/93.6 | 62.1/90.8 | 41.8/97.6 | 61.6/93.6 | 61.2/88.8 | 97.0/21.8 | 99.6/1.5 | 98.3/11.6 |
| **CC-DIsoN** | 75.4/78.8 | 79.5/61.7 | 72.6/79.7 | 63.0/89.4 | 64.0/85.3 | 70.7/79.9 | 61.8/92.0 | 69.6/81.0 | 98.3/4.8 | 98.4/4.0 | 98.3/4.4 |

## 4.1 Evaluation on Medical OOD Benchmarks

**Dermatology, Chest X-Ray and Breast Ultrasound.** Tab. 1 compares CC-DIsoN with the baselines on our three medical OOD benchmarks where the OOD task is to detect artifacts. We can see that the post-hoc methods struggle to detect these domain-specific artifacts: fDBD and ViM average only between $63 - 64\%$ AUROC while having a high FPR95 ($84.6\%$ and $78.1\%$, respectively). Training-time regularization methods like CIDER do better ($70.2\%$ average AUROC), but still have a high FPR95 of $72.8\%$. CC-DIsoN performs strongly compared to the baselines and shows consistent improvement across all datasets. Compared to fDBD, it improves AUROC by **15.8%** and reduces FPR95 by **25.4%**. Against the best baseline (CIDER), it improves AUROC by **9.8%** and lowers FPR95 by **13.6%**. This demonstrates that directly comparing test samples against the training data during inference has a positive impact on OOD detection.

**Effects of Class-Conditioning.** Tab. 3a shows the benefits of class-conditioning (Sec. 3.3). On average, CC-DIsoN improves AUROC by 1.7%. More notably, CC-DIsoN reduces FPR95 by **8.2%** across all three datasets. Class-conditioning consistently lowers FPR95, demonstrating that focusing the isolation task on the predicted same-class samples improves OOD detection.

Table 3: (a) Comparison of DIsoN and CC-DIsoN on three medical datasets using AUROC (higher is better) and FPR95 (lower is better). (b) Effect of incorrect predicted classes for class-conditional sampling in CC-DIsoN (AUROC). "Best Baseline" is the strongest baseline from Tab. 1 (for reference); "ID & OOD wrong": all targets assigned incorrect classes; "ID wrong": only ID targets assigned incorrect classes; "OOD wrong": only OOD targets assigned incorrect classes.

(a)

| Dataset | AUROC $\uparrow$ | | FPR95 $\downarrow$ | |
|---|---|---|---|---|
| | DIsoN | CC-DIsoN | DIsoN | CC-DIsoN |
| Ultrasound | **67.0** | 65.6 | 78.6 | **73.2** |
| Dermatology | 86.4 | **89.5** | 50.0 | **42.5** |
| X-Ray | 81.4 | **84.9** | 73.6 | **61.9** |
| Average | 78.3 | 80.0 | 67.4 | 59.2 |

(b)

| Dataset | Best Baseline | Standard CC-DIsoN | ID & OOD wrong | ID wrong | OOD wrong |
|---|---|---|---|---|---|
| Ultrasound | 61.0 | 65.6 | 61.5 | 34.6 | 89.6 |
| Dermatology | 82.0 | 89.5 | 83.1 | 84.7 | 86.7 |
| X-Ray | 70.5 | 84.9 | 77.9 | 74.0 | 88.3 |

**Impact of Predicted Class Accuracy.** Since CC-DIsoN conditions on the predicted class from the pre-trained model $M_{pre}$, we analyse how misclassification affect its performance. We simulate three

controlled scenarios by asssigning wrong predicted classes to: (i) all target samples (ID & OOD), (ii) only ID targets while OOD targets use their predicted classes, and (iii) only OOD targets while ID targets use their predicted classes. Results in Tab.3b show that performance drops mainly when ID targets are incorrect, as class-conditioning compares them to unrelated classes, making isolation easier and reducing the separation in convergence rounds between ID and OOD samples. In contrast, performance slightly increases when OOD targets are mislabeled, as comparing them to unrelated classes makes convergence faster. Even under such extreme settings, CC-DIsoN remains competitive. Further details are provided in the Appendix E.

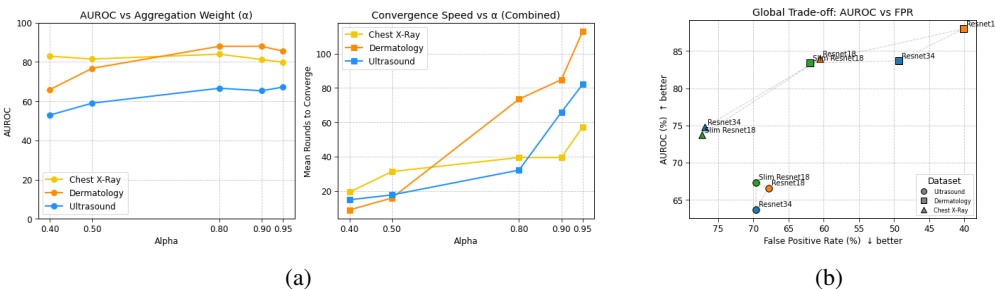

(a)            (b)

Figure 4: **(a) Effect of aggregation weight** $\alpha$. Left: AUROC vs. $\alpha$. Higher $\alpha$ improves OOD detection by emphasizing the source updates. Right: Mean number of communication rounds until convergence (ID and OOD targets combined). Trade-off: Lower $\alpha$ values speed up convergence but reduce OOD performance. **(b) Network Size.** Global AUROC vs. FPR95 plot for three backbones (Slim ResNet18, ResNet18, ResNet34) across the same datasets. ResNet18 gives the balance between AUROC and FPR95. Grey dashed lines link backbones per dataset.

**Histopathology: MIDOG.** While earlier benchmarks focused on detecting non-diagnostic artifacts as OOD task, MIDOG evaluates semantic and covariate shift as OOD task. Tab. 2 shows that CC-DIsoN also performs strongly in this setting. In the most challenging near-OOD setting, CC-DIsoN achieves an average AUROC of 69.6%, which is a **8.4%** improvement compared to the best regularization-based method PALM and a **7.2%** gain compared to the best post-hoc method ViM. It also achieves the lowest FPR95 in this setting, albeit with a smaller margin. In the less challenging far-OOD setting, CC-DIsoN achieves an AUROC of 98.3% and a FPR95 of 4.4%, indicating almost perfect separation. Out of all compared methods, only PALM achieves same AUROC performance in the far-OOD task, although CC-DIsoN performs better in the near-OOD task. IForests were originally developed for tabular data, where they perform well. However, prior works [34, 10] shows poor performance on high-dimensional imaging data, which explains the low AUROC in our experiments. Overall, these results demonstrate that CC-DIsoN does not only perform well for OOD tasks with localised artifacts but also effective in identifying semantic and covariate shifts across medical imaging tasks.

### 4.2 Further Analysis of DIsoN

**Sensitivity Analysis of Hyperparameter** $\alpha$. We analyze the effect of the aggregation weight $\alpha$ (Eq. 2) on both the OOD detection performance and convergence rate across Dermatology, Ultrasound, and X-Ray in Fig. 4a. We can see that lower $\alpha$ values, which give relatively more weight to the target updates, reduce OOD detection performance, since the isolation task becomes dominated by the target sample, and loses the comparison signal to the source data. Increasing $\alpha$ improves AUROC consistently across datasets, with performance plateauing at $\alpha = 0.8$. This shows that a stronger emphasis on training data improves the isolation-based OOD performance. However, the number of communication rounds required for convergence increases with $\alpha$. This aligns with Proposition 3.1, where $\alpha$ controls the implicit oversampling ratio N: smaller $\alpha$ increases the target signal (larger N) (faster isolation), while larger $\alpha$ slows convergence (more rounds needed to incorporate the target's signal). In practice, this creates a trade-off between convergence speed and detection performance. The goal of this analysis is not to identify one "optimal" value of $\alpha$, but to show that our method remains robust across a wide range of values. Performance stays quiet stable for $\alpha$ between 0.5 and 0.95 (Fig. 4a), suggesting good generalization across tasks without extensive hyperparameter tuning.

**Effect of Network Size.** To quantify how capacity of the Isolation Network affects the OOD scores, we compare three network sizes: a "Slim" ResNet18 (0.5× channel widths), the standard

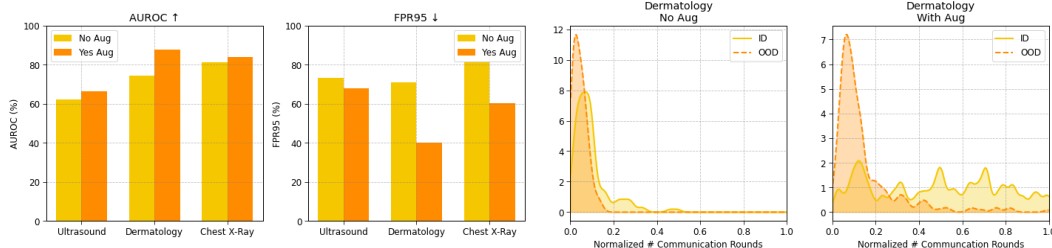

Figure 5: **Effect of image augmentation.** Left: Bar plots show that using random augmentation ("Yes Aug") during training improves AUROC and FPR95 for three datasets. Right: Density plots for dermatology: without augmentation, ID and OOD curves overlap heavily, whereas with augmentation OOD samples isolate fast and ID samples require more updates.

ResNet18, and a deeper ResNet34. Fig. 4b shows each backbone's AUROC vs. FPR95 trade-off on Breast Ultrasound (circles), Dermatology (squares), and Chest X-Ray (triangles). Neither the Slim-ResNet18 nor the ResNet34 outperforms the standard ResNet18: the slim model might lack sufficient representational capacity to detect the subtle visual difference, while the deeper network does not yield consistent OOD gains and roughly doubling the required compute. This is likely because a binary isolation task does not require this excessive parameter capacity. Therefore, ResNet18 gives the best balance between good ID/OOD separation and computational efficiency.

**Effect of Image Augmentations.** Fig. 5 shows the effects of applying image augmentations during DIsoN training on Ultrasound, Dermatology, and X-Ray. Augmentations consistently improves AUROC: **+4.43%** on Ultrasound, **+13.69%** on Dermatology, and **+2.76%** on X-ray. FPR95 also decreases, most notably on Dermatology **-30.99%**. The density plot of dermatology dataset shows that without augmentation, ID and OOD samples converge in similar number of few rounds, resulting in poor separation. With augmentations, OOD target samples isolate quickly, but ID target samples require many more rounds. This demonstrates the regularization effect of augmentations: they prevent the model from quickly memorizing a single target sample, regardless of whether it is ID or OOD.

**Runtime Analysis and Practical Considerations.** We extend the runtime analysis in Fig. 4a by reporting detailed statistics, including quantiles and results across different $\alpha$ values, in Appendix G. Since wall-clock time depends on hardware and network conditions, we primarily measure runtime in communication rounds, offering a hardware-independent estimate. Although DIsoN introduces extra computation, as each target sample requires an isolation task, this design trades speed for improved OOD detection performance. In practice, real-time inference is rarely required in healthcare, where scans are often reviewed hours or days later [31] (except in emergencies), and diagnostic accuracy is typically prioritised over speed. Further runtime comparisons with baselines, as well as notes on parallelisation and more detailed practical considerations, are provided in Appendix H.

## 5 Conclusion

In this paper, we propose Decentralized Isolation Networks (DIsoN), a novel OOD detection framework that, unlike most existing methods, actively leverages training data at inference, without requiring data sharing. DIsoN trains a binary classification task to measures the difficulty of isolating a test sample by comparing it to the training data through model parameter exchange between the source and deployment site. Our class-conditional variant, CC-DIsoN, further improves performance and achieves consistent gains in AUROC and FPR95 across four medical imaging datasets and 12 OOD detection tasks, compared to state-of-the-art methods. One limitation of DIsoN is, that it requires additional compute during inference for target samples. In practice, DIsoN requires roughly 40s to 4 min per sample (depending on the dataset), thus it is practical for applications where inference delay of this magnitude is not an issue. We show that this overhead can be controlled via the aggregation weight $\alpha$ (convergence speed) and backbone size, enabling a trade-off between efficiency and detection performance. In future work, we aim to extend DIsoN to handle multiple target samples simultaneously to improve efficiency. Overall, our results demonstrate that leveraging training data during inference can improve OOD detection in privacy-sensitive deployment scenarios.

## Acknowledgements

FW is supported by the EPSRC Centre for Doctoral Training in Health Data Science (EP/S02428X/1), the Anglo-Austrian Society, and an Oxford-Reuben scholarship. PS is supported by the EPSRC Programme Grant [EP/T028572/1], UKRI grant [EP/X040186/1], and EPSRC Doctoral Training Partnership award. HA is supported by a scholarship via the EPSRC Doctoral Training Partnerships programme [EP/W524311/1, EP/T517811/1]. The authors acknowledge the use of the University of Oxford Advanced Research Computing (ARC) facility (http://dx.doi.org/10.5281/zenodo.22558).

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

# A  DIsoN and CC-DIsoN: Algorithm

In this section, we present pseudocode for our DIsoN method and its class-conditional variant, CC-DIsoN. Algorithm 1 describes the code that is executed on the Source Node, while Algorithm 2 runs on the Target Node. Lines specific to CC-DIsoN are highlighted in blue. The **initialization step** occurs in lines 4–5 of Algorithm 1 and line 5 of Algorithm 2. The **local updates** step is performed on lines 7–12 (Source) and 7–8 (Target), followed by the **aggregation step** on line 14 on the Source Node. If the convergence criteria (Def. 3.1) are not met, another communication round starts.

The algorithms demonstrate that implementing CC-DIsoN requires only minor changes compared to DIsoN: before initialization, the Target Node predicts the target sample's class using the pre-trained model and sends it to the Source Node (lines 2–4 of Algorithm 2). The Source Node then filters its training data to sample batches from the predicted class *only* (line 9 of Algorithm 1).

---

**Algorithm 1** Source Node: DIsoN / CC-DIsoN

1: **function** SOURCENODE($\mathcal{D}_s$, $\theta^{\mathrm{pre}}$)
2:     **if** CC-DIsoN :
3:         **receive** $\hat{y}$ from target
4:     $\theta^f \leftarrow \theta^{\mathrm{pre}}$; $\theta^h \leftarrow \mathrm{rand}$; $\theta^{(0)} \leftarrow (\theta^f, \theta^h)$         ▷ initialize global model
5:     **send** $\theta^{(0)}$ to Target; $\theta_S \leftarrow \theta^{(0)}$
6:     **for** $r = 1$ **to** $R$ **do**         ▷ communication rounds
7:         **for** $e = 1$ **to** $E$ **do**         ▷ local updates on Source
8:             **if** CC-DIsoN :
9:                 $B_s \sim \{(\mathbf{x}_s, y_s) \in \mathcal{D}_s \mid y_s = \hat{y}\}$     ▷ filter $B_s$ on predicted $\mathbf{x}_t$ class
10:             **else**
11:                 sample $B_s \sim \mathcal{D}_s$
12:             $\theta_S \leftarrow \theta_S - \eta \nabla_{\theta_S} \left[ \frac{1}{|B_s|} \sum_{\mathbf{x}_s \in B_s} L(\theta_S; \mathbf{x}_s, 0) \right]$
13:         **receive** $\theta_T$ from Target
14:         **aggregation:**   $\theta^{(r)} \leftarrow \alpha \theta_S + (1 - \alpha) \theta_T$         ▷ Eq. 2
15:         SourceConverged $\leftarrow$ converged($\theta^{(r)}, \mathcal{D}_s$)     ▷ Test criteria 2 (Def. 3.1)
16:         **send** $\theta^{(r)}$ and SourceConverged to Target
17:         $\theta_S \leftarrow \theta^{(r)}$     ▷ update Source model for next comm. round

---

**Algorithm 2** Target Node: DIsoN / CC-DIsoN

1: **function** TARGETNODE($x_t$, $M_{\mathrm{pre}}$, $R$)
2:     **if** CC-DIsoN :
3:         $\hat{y} \leftarrow \arg\max_c [M_{\mathrm{pre}}(\mathbf{x}_t)]_c$
4:         **send** $\hat{y}$ to source
5:     **receive** $\theta^{(0)}$ from Source; $\theta_T \leftarrow \theta^{(0)}$     ▷ init. Target model with global model
6:     **for** $r = 1$ **to** $R$ **do**         ▷ communication rounds
7:         **for** $e = 1$ **to** $E$ **do**         ▷ local updates on Target
8:             $\theta_T \leftarrow \theta_T - \eta \nabla_{\theta_T} L(\theta_T; \mathbf{x}_t, 1)$
9:         **send** $\theta_T$ to Source
10:         **receive** updated $\theta^{(r)}$ and SourceConverged from Source
11:         **if** converged($\theta^{(r)}, \mathbf{x}_t$) **and** SourceConverged :   ▷ Test crit. 1 & 2 (Def. 3.1)
12:             **break**
13:         $\theta_T \leftarrow \theta^{(r)}$     ▷ update Target model for next comm. round
14:     **return** $S_{DIsoN}(\mathbf{x}_t) = r$

---

# B  Connection between DIsoN and Isolation Network: Proof Proposition 3.1

This section provides the full proof of Proposition 3.1, which states that if we set $E = 1$ and the aggregation weights in Eq. 2 accordingly, DIsoN becomes equivalent to the centralized Isolation

Network. This result also demonstrates how the aggregation weight $\alpha$ implicitly controls the number of times the target sample is oversampled ($N$) in Eq. 1 for the Isolation Network.

We begin by restating Proposition 3.1:

**Proposition 3.1** (DIsoN and Centralized Isolation Network Equivalence for $E = 1$). *Let $\theta_{cent}$ be the model parameters from our centralized algorithm and $\theta_{dec}$ be the parameters from DIsoN. Let each site perform one local SGD step ($E = 1$) with learning rate $\eta$, and aggregate with $\alpha = \frac{|B_s|}{|B_s|+N}$, $\beta = \frac{N}{|B_s|+N}$. Then the decentralized update equals the centralized one:*

$$\theta_{\text{dec}}^{(r+1)} = \theta^{(r)} - \eta\big(\alpha\, g_S(\theta^{(r)}) + \beta\, g_T(\theta^{(r)})\big) = \theta^{(r)} - \eta\, g_c(\theta^{(r)}) = \theta_{\text{cent}}^{(r+1)},$$

*where $g_S(\theta) = \frac{1}{|B_s|}\sum_{\mathbf{x}_s \in B_s}\nabla_\theta L(\theta; \mathbf{x}_s, 0)$, $g_T(\theta) = \nabla_\theta L(\theta; \mathbf{x}_t, 1)$ and $N$ is the number of times $\mathbf{x}_t$ is oversampled in the centralized version.*

*Proof.* Recall that $B_s$ is a mini-batch of $|B_s|$ source samples and let the target sample $\mathbf{x}_t$ be oversampled $N$ times in the Isolation Network. The local gradient on the source node $g_S$ and on the target node $g_T$ are defined as:

$$g_S(\theta) = \frac{1}{|B_s|}\sum_{\mathbf{x}_s \in B_s}\nabla_\theta L(\theta; \mathbf{x}_s, 0), \qquad g_T(\theta) = \nabla_\theta L(\theta; \mathbf{x}_t, 1),$$

and the *centralized* mini-batch gradient of the Isolation Network is defined as:

$$g_c(\theta) = \frac{1}{|B_s| + N}\Big(\sum_{\mathbf{x}_s \in B_s}\nabla_\theta L(\theta; \mathbf{x}_s, 0) + N \cdot \nabla_\theta L(\theta; \mathbf{x}_t, 1)\Big). \tag{A.1}$$

**Step 1. Local parameter updates on the Source Node and Target Node.** After one SGD step ($E = 1$) with learning rate $\eta$ we have,

$$\theta_S^{(r,1)} = \theta^{(r)} - \eta\, g_S(\theta^{(r)}), \qquad \theta_T^{(r,1)} = \theta^{(r)} - \eta\, g_T(\theta^{(r)}). \tag{A.2}$$

**Step 2. Weighted aggregation.** Using the aggregation Eq. 2 with $\alpha = \dfrac{|B_s|}{|B_s| + N}$, $\beta = \dfrac{N}{|B_s| + N}$, $\beta = 1 - \alpha$ gives

$$\begin{aligned}
\theta_{\text{dec}}^{(r+1)} &= \alpha\, \theta_S^{(r,1)} + \beta\, \theta_T^{(r,1)} \\
&= \alpha\big(\theta^{(r)} - \eta\, g_S\big) + \beta\big(\theta^{(r)} - \eta\, g_T\big) \quad \text{(by (A.2))} \\
&= (\alpha + \beta)\theta^{(r)} - \eta\big(\alpha\, g_S + \beta\, g_T\big) \\
&= \theta^{(r)} - \eta\big(\alpha\, g_S + \beta\, g_T\big).
\end{aligned} \tag{A.3}$$

**Step 3. Centralized gradient of the Isolation Network.**

$$\alpha\, g_S = \frac{|B_s|}{|B_s| + N}\Big(\frac{1}{|B_s|}\sum_{\mathbf{x}_s \in B_s}\nabla_\theta L(\theta; \mathbf{x}_s, 0)\Big) = \frac{1}{|B_s| + N}\sum_{\mathbf{x}_s \in B_s}\nabla_\theta L(\theta; \mathbf{x}_s, 0),$$

$$\beta\, g_T = \frac{N}{|B_s| + N}\nabla_\theta L(\theta; \mathbf{x}_t, 1).$$

Adding the two terms reproduces (A.1):

$$\alpha\, g_S + \beta\, g_T = \frac{1}{|B_s| + N}\Big(\sum_{\mathbf{x}_s \in B_s}\nabla_\theta L(\theta; \mathbf{x}_s, 0) + N\nabla_\theta L(\theta; \mathbf{x}_t, 1)\Big) = g_c(\theta). \tag{A.4}$$

**Step 4. Equality of parameter updates.** Substituting (A.4) into (A.3) results in

$$\theta_{\text{dec}}^{(r+1)} = \theta^{(r)} - \eta\, g_c(\theta^{(r)}) = \theta_{\text{cent}}^{(r+1)}, \tag{A.5}$$

which is exactly the centralized Isolation Network SGD update. This shows that the decentralized and centralized updates are equivalent when $E = 1$ and the aggregation weights are chosen as $\alpha = \frac{|B_s|}{|B_s|+N}, \beta = \frac{N}{|B_s|+N}$.

$\square$

Table 4: Class-wise dataset splits used in our experiments, showing the total number of ID images per class, splits into pre-training and ID test sets, OOD detection task, number of OOD test samples, and image resolution. For Histopathology, we report near-OOD (domains 2–7) and far-OOD (CCAgT, FNAC 2019) tasks separately.

| Dataset | Class | Total: # ID Images | Pre-train: # Images | # ID Test | OOD Task | # OOD Test | Img. Size |
|---|---|---|---|---|---|---|---|
| Dermatology | Nevus | 832 | 745 | 87 | black ruler | 251 | 224×224 |
| | Not Nevus | 571 | 516 | 55 | | | |
| Ultrasound | Normal | 126 | 114 | 12 | text annotations | 228 | 224×224 |
| | Benign | 269 | 245 | 24 | | | |
| | Malignant | 157 | 137 | 20 | | | |
| Chest X-Ray | Cardiomegaly | 1788 | 1711 | 77 | pacemaker | 1000 | 224×224 |
| | No Cardiomegaly | 21557 | 20634 | 923 | | | |
| Histopathology | Mitotic | 421 | 375 | 46 | near-OOD: domains 2–7 | 500 | 50×50 |
| | Imposter | 663 | 581 | 82 | far-OOD: CCAgT & FNAC | per domain | |
| | Neither | 1063 | 940 | 123 | | | |

# C   Dataset Details.

We evaluate DIsoN on four medical imaging datasets with two main OOD settings: (i) **artifact detection** (Dermatology, Breast Ultrasound, Chest X-ray) and (ii) **semantic/covariate shift detection** (Histopathology). In this section, we provide more detailed dataset information with class-wise splits used in our experiments. Table 4 reports the total number of images per class, splits into pre-training and ID test samples, the number of OOD test samples, the OOD task and the image size.

For Dermatology (1403 artifact-free ID images) and Breast Ultrasound (552 artifact-free ID images), we follow the benchmark setup from [3], using manually annotated artifact-free images for pre-training and ID testing, and ruler/text annotation artifacts as OOD. The Chest X-ray dataset uses 23,345 frontal-view scans without support devices as ID data, following the setup from [2], with scans with pacemakers as OOD artifacts. In all three datasets, the ID data is split 90/10 into pre-training for the main task of interest and ID test sets.

For Histopathology, we follow the recently published OpenMIBOOD [11] benchmark, which uses a similar setup to the well-known OpenOOD benchmark [42], but is specialized on medical images. OpenMIBOOD demonstrates that many state-of-the-art OOD detection methods fail to generalize to medical data. The dataset is split into multiple domains. The dataset consists of Hematoxylin & Eosin–stained histology patches grouped into multiple domains. Following the protocol in [11], we use the training split of domain 1a (1,896 ID images) for pre-training and its test split (251 images) as the ID evaluation set.

Domains 2–7 are treated as near-OOD and include seven distinct cancer types (breast carcinoma, lung carcinoma, lymphosarcoma, cutaneous mast cell tumor, neuroendocrine tumor, soft tissue sarcoma, and melanoma), from both human and canine. These domains introduce semantic shifts in cell types as well as covariate shifts due to different staining protocols and imaging hardware. For far-OOD, two external datasets are used, that introduce a strong semantic shift: CCAgT [1], which uses AgNOR staining, and FNAC 2019 [32], which uses Pap staining, both differ significantly from the Hematoxylin & Eosin staining used for ID data and are also used for different medical applications.

# D   Training Details & Hyperparameters

In addition to the training details provided in Section 4, this section presents further training details and hyperparameters.

Table 5: **Pre-training hyperparameters.** These settings are used to train the main classification task before initializing DIsoN. LR: learning rate; BS: batch size.

| Dataset | Main Task | Arch. | Optim. | Epochs | LR | BS |
|---|---|---|---|---|---|---|
| Dermatology | Nevus vs. Non-Nevus | ResNet18 | Adam | 750 | $1 \times 10^{-3}$ | 32 |
| Breast Ultrasound | Normal / Benign / Malignant | ResNet18 | Adam | 1000 | $1 \times 10^{-3}$ | 32 |
| Chest X-Ray | Cardiomegaly vs. No Cardiomegaly | ResNet18 | Adam | 500 | $1 \times 10^{-3}$ | 32 |
| Histopathology | Mitotic / Imposter / Neither | ResNet18 | SGD (0.9 momentum) | 300 | $5 \times 10^{-4}$ | 128 |

Table 6: **Training hyperparameters used for DIsoN experiments.** Overview of architecture, optimizer, learning rate (LR), and batch size (BS) used to train DIsoN for each dataset.

| Dataset | Arch. | Optim. | LR | BS |
|---|---|---|---|---|
| Dermatology | ResNet18 | Adam | $1 \times 10^{-3}$ | 16 |
| Breast Ultrasound | ResNet18 | Adam | $1 \times 10^{-3}$ | 8 |
| Chest X-Ray | ResNet18 | Adam | $3 \times 10^{-3}$ | 16 |
| Histopathology | ResNet18 | Adam | $1 \times 10^{-2}$ | 16 |

## D.1 Pre-Training for the Main Task of Interest

We pre-train a ResNet18 on each dataset's main classification task. DIsoN models use Instance Normalization, as described in Section 3. For baseline methods, we use Batch Normalization (BN) to ensure a fair comparison, since they assume BN in their original setups. Table 5 summarizes the training settings, using the dataset split in the "Pre-train" column of Table 4. For the Histopathology dataset, we follow the protocol from [11], initializing with ImageNet-1k [8] pre-trained weights to reduce training time, and use SGD with momentum. All other models are trained from scratch using Adam.

## D.2 Training DIsoN

This subsection provides a more detailed description of the DIsoN training setup introduced in Section 4. Table 6 summarizes the training hyperparameters. The Source Node uses the pre-training split from Table 4 as its training data. As described earlier, we set the number of local iterations per communication round such that it approximately matches one epoch over the training data. To limit runtime, we also use a maximum number of communication rounds for each dataset. If convergence is not reached within this limit, we assign the maximum round $R_{\max}$ as the OOD score. We use $R_{\max} = 300$ for Dermatology and Ultrasound, and $R_{\max} = 100$ for the longer-running Chest X-ray and Histopathology datasets.

**Image Augmentations.** As described in Section 3, we apply standard stochastic image augmentations across all datasets. The augmentations for each dataset are mostly identical, with only minor dataset-specific adjustments. For Histopathology, due to the smaller image size, we reduce the rotation range and leave out random cropping. For Breast Ultrasound, we replace random cropping with color jitter. The full set of augmentations are listed below:

- **Random rotation:** $\pm 15°$ (use $\pm 5°$ for Histopathology)
- **Random crop:** $224 \times 224$ with padding=25 (applied to Chest X-Ray & Dermatology)
- **Color jitter:** brightness=0.1, contrast=0.1 (applied to Breast Ultrasound)

**Convergence Parameter Choices for $E_{\text{stab}}$ and $\tau$.** The parameters $E_{\text{stab}}$ and $\tau$ were chosen to capture the convergence behavior of the isolation process. Specifically, $E_{\text{stab}}$ acts as a patience parameter, conceptually similar to early-stopping criteria in learning-rate schedulers, requiring correct classification of $\mathbf{x}_t$ for several consecutive rounds to ensure stable convergence. We set $E_{\text{stab}} = 5$, a commonly used value in schedulers that effectively captures convergence. The confidence threshold $\tau = 0.85$ ensures that the model learned to separate the target sample with sufficient confidence. This value follows standard practice in uncertainty-based literature [18]. In preliminary experiments, these values consistently produced stable and reliable results, so they were kept fixed across all datasets to avoid dataset-specific tuning or overfitting.

**Effect of partial Fine-Tuning** We also examined whether freezing parts of the network could be beneficial for the isolation task by simplifying optimization. Freezing the backbone, however, assumes that the pre-trained feature extractor already provides sufficiently expressive embeddings to distinguish ID from OOD samples. We found this assumption too restrictive: in early experiments, partial fine-tuning of the network (e.g., only the head or last block) led to worse OOD detection performance than fine-tuning the entire model.

# E  Additional Details: Ablation Study for Impact of Predicted Class Accuracies for Class-Conditional Sampling

CC-DIsoN uses predictions of a primary model $M_{\mathrm{pre}}$, deployed for a task of interest (e.g., cardiomegaly classification in X-rays; see Sec. 4), for class-conditional sampling. This enables comparing a target sample to the most relevant subset of training data (its predicted class) rather than to all training samples. Our ablations on the effects of class-conditioning (Tab. 3a) demonstrated that class-conditional sampling improves OOD detection performance, while DIsoN without class-conditioning still performs well (see comparison between Tabs. 3a and 1).

To further analyze the influence of class-conditioning and how misclassifications affect the results, we provide here a more detailed analysis of the ablation study on predicted class accuracy introduced in Section 4.1. We first measure the classification accuracy of $M_{\mathrm{pre}}$ on ID, OOD, and combined target samples, along with the corresponding CC-DIsoN AUROC scores (Tab. 7). Typically, especially in safety-critical domains such as healthcare, high ID accuracy is a requirement for deployment. It is also expected that classification accuracy decreases on OOD samples due to domain shift. Comparing these accuracies with OOD detection performance shows no clear correlation, for example, the Dermatology dataset achieves the highest AUROC (89.5) despite the lowest classification accuracy (63.0%). This indicates that CC-DIsoN's OOD detection performance does not depend on perfectly accurate predicted classes but rather on the data characteristics.

Table 7: Classification accuracy of the primary models on ID, OOD, and combined target samples, together with CC-DIsoN OOD detection performance (AUROC from Tab. 1).

| Dataset | ID Acc. (%) | OOD Acc. (%) | ID&OOD Acc. (%) | CC-DIsoN AUROC |
|---|---|---|---|---|
| Dermatology | 77.6 | 54.0 | 63.0 | 89.5 |
| Chest X-Ray | 93.0 | 70.8 | 82.2 | 84.9 |
| Ultrasound | 78.6 | 61.4 | 64.8 | 65.6 |
| MIDOG (near+far) | 77.3 | 68.6 | 69.2 | 76.0 |

In Section 4.1 and Tab. 3b, we further investigate the effect of incorrect predicted classes on class-conditioning in controlled settings. Here, we describe these settings in more detail. We designed three experiment variants for Dermatology, Breast Ultrasound, and Chest X-ray by manually assigning wrong predicted classes from the primary model to specific target samples:

1. **All targets mislabeled (ID + OOD wrong):** All target samples, regardless of whether they are ID or OOD, are assigned a random incorrect class. This represents the extreme case of a completely inaccurate primary model (0% Accuracy), which would not be realistic in deployment but an interesting setting to obtain insights.

2. **Only ID targets mislabeled:** All target ID samples are assigned incorrect classes (0% Accuracy on ID), while OOD targets use their predicted classes. Again, this represents an impractical scenario for deployment, since a model with 0% ID accuracy would not be used in real-world, but useful to study misclassification behavior. Central motivation for class-conditioned sampling was to make the isolation task more difficult for ID target samples, as they should be harder to isolate from source (training ID) samples of the same class, as they share common visual features. This should lead to slower convergence and increase the difference than when separating OOD samples from source (training ID) samples, even of the same class, which is faster due to domain-shift. If we instead compare ID target samples to source (training) samples from the wrong class, the isolation will be easier, resulting in worse OOD detection.

3. **Only OOD targets mislabeled:** All OOD target samples are assigned incorrect classes (0% Accuracy on OOD), while ID targets use their predicted classes. This mimics a situation where domain shift causes the primary model to misclassify OOD samples, and gives us insights what would happen in the most extreme cases of domain shift. Since OOD data already differ visually from the source distribution, assigning and comparing them with unrelated classes can make their isolation even easier, and we expect an improvment in detection.

The results in Tab. 3b confirm these expectations. Performance decreases when ID targets are mislabeled, where class-conditioning is most important, while slightly increasing when OOD targets are mislabeled. Even in the extreme "all wrong" case, CC-DIsoN remains competitive and outperforms the strongest baseline (Tab. 3b) in all but one dataset. This shows that CC-DIsoN's improvement does not rely on perfectly accurate class predictions and that the method remains robust to moderate number of misclassification errors. Overall, comparison with visually similar source samples is beneficial for DIsoN and provides stable OOD detection performance even when predicted classes are not perfect.

## F  Ablation Study: Further Investigation of Image Augmentations

In Section 4.2, we showed that applying image augmentations during DIsoN training improves OOD detection across Dermatology, Chest X-ray, and Breast Ultrasound. Here, we extend this ablation study by comparing four augmentation settings: no augmentation, augmentations only on the Source Node data, only on the target sample, and on both nodes. Fig. 6 reports AUROC and FPR95 for all settings. Applying augmentations on both nodes consistently performs best. Interestingly, for Dermatology, applying augmentations only on the target sample gives best AUROC overall. We can also see that target-only augmentation outperforms source-only augmentations in most cases. One exception is Ultrasound FPR95, where source-only yields better results. These findings further strengthen our finding of the importance of regularization on especially the target sample to prevent fast memorization of superficial image-specific features, and encourage learning more meaningful features for differentiation of ID and OOD data, as discussed in Section 4.2.

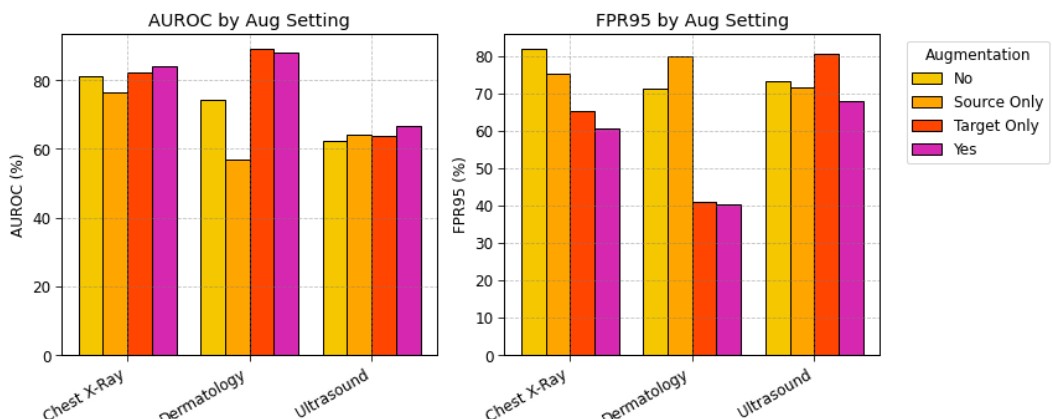

Figure 6: **Effect of applying augmentations on different nodes for DIsoN.** AUROC (left) and FPR95 (right) across three medical datasets under four augmentation settings: no augmentation, augmentations only on the source node, only on the target sample, and on both. Applying augmentations on both nodes performs best overall. Target-only augmentation outperforms source-only in most settings, highlighting the importance of regularizing the target sample during isolation.

## G  Runtime Statistics

Figure 4 (a) in the main paper reports the average number of rounds required for the isolation task to converge. To provide a more detailed view of runtime variability, we report additional statistics in Table 8, including the 25th percentile, median, and 75th percentile of convergence rounds for both ID and OOD target samples using our default $\alpha = 0.8$. Reporting these quantiles helps practitioners assess expected runtime under different ID/OOD ratios. Since wall-clock runtime depends on hardware, we report the number of rounds as a hardware-independent measure. OOD samples generally converge in fewer rounds than ID samples, consistent with their easier separability from the source distribution.

We further analyze how the runtime distribution changes with $\alpha$. Tables 9-11 show that decreasing $\alpha$ consistently reduces the number of rounds required for convergence. Importantly, OOD samples

Table 8: Quantiles (25th, median, 75th percentile) of convergence rounds for ID and OOD samples across datasets. OOD samples converge in fewer rounds and show lower variability, while ID samples display higher variability due to within-distribution complexity.

| Dataset | Sample Type | 25th Perc. | Median | 75th Perc. |
|---|---|---|---|---|
| Dermatology | ID | 68.0 | 126.5 | 184.0 |
| | OOD | 22.0 | 28.0 | 40.0 |
| Ultrasound | ID | 20.0 | 29.0 | 100.0 |
| | OOD | 16.0 | 20.5 | 32.0 |
| Chest X-Ray | ID | 28.0 | 41.0 | 92.0 |
| | OOD | 16.0 | 20.0 | 26.0 |
| Histopathology (near OOD) | ID | 44.3 | 59.5 | 74.0 |
| | OOD | 34.0 | 44.0 | 57.0 |

converge faster across all $\alpha$, while ID samples exhibit broader variability due to within-distribution complexity. Note that, as described above, we set the maximum number of rounds $R_{max}$ as the OOD score when convergence was not reached. For these additional runtime experiments, we set a lower maximum number of rounds for the Dermatology dataset from ($R_{max} = 300$) to ($R_{max} = 150$) in order to reduce computational cost. For $\alpha = 0.90$ and $\alpha = 0.95$, the majority of ID samples did not converge before reaching this limit. Therefore, the 25th percentile equals $R_{max}$ in those cases.

Table 9: Distribution of convergence rounds (25th, median, 75th percentile) for Breast Ultrasound across $\alpha$ values. Lower $\alpha$ accelerates convergence, while OOD samples consistently require fewer rounds than ID samples.

| $\alpha$ | Sample Type | 25th Perc. | Median | 75th Perc. |
|---|---|---|---|---|
| 0.95 | ID | 39.8 | 50.0 | 300.0 |
| | OOD | 33.0 | 42.0 | 59.0 |
| 0.90 | ID | 25.0 | 39.5 | 297.0 |
| | OOD | 22.0 | 27.0 | 42.0 |
| 0.50 | ID | 13.8 | 17.0 | 26.3 |
| | OOD | 13.0 | 15.0 | 19.0 |
| 0.40 | ID | 11.0 | 13.0 | 17.0 |
| | OOD | 11.0 | 14.0 | 17.0 |

Table 10: Distribution of convergence rounds (25th, median, 75th percentile) for Chest X-Ray across $\alpha$ values. Lower $\alpha$ reduces the number of rounds required for convergence, with OOD samples converging faster and more consistently than ID samples.

| $\alpha$ | Sample Type | 25th Perc. | Median | 75th Perc. |
|---|---|---|---|---|
| 0.95 | ID | 51.0 | 58.0 | 75.0 |
| | OOD | 44.0 | 48.0 | 53.0 |
| 0.90 | ID | 26.0 | 35.0 | 100.0 |
| | OOD | 19.0 | 23.0 | 27.0 |
| 0.50 | ID | 20.0 | 27.0 | 51.5 |
| | OOD | 14.0 | 16.0 | 20.0 |
| 0.40 | ID | 15.0 | 17.0 | 23.0 |
| | OOD | 11.0 | 13.0 | 15.0 |

## H  Practical Deployment Considerations And Runtime Comparisons

In many medical imaging workflows, real-time inference is not required. Diagnostic scans are often reviewed hours or days after acquisition due to limited clinical staff availability, and in some cases, turnaround times can extend to several weeks [31]. Within such workflows, AI models can operate asynchronously, processing scans during this waiting period. Therefore, inference times of several minutes per sample are acceptable when they provide more reliable detection of OOD samples, and similar latency is acceptable in many other non-real-time workflows outside of healthcare. DIsoN was

Table 11: Distribution of convergence rounds (25th, median, 75th percentile) for Dermatology across $\alpha$ values. Convergence becomes substantially faster as $\alpha$ decreases, and OOD samples consistently require fewer rounds than ID samples.

| $\alpha$ | Sample Type | 25th Perc. | Median | 75th Perc. |
|---|---|---|---|---|
| 0.95 | ID | 150.0 | 150.0 | 150.0 |
| | OOD | 72.0 | 87.0 | 122.0 |
| 0.90 | ID | 150.0 | 150.0 | 150.0 |
| | OOD | 37.0 | 44.0 | 66.0 |
| 0.50 | ID | 16.0 | 17.0 | 21.0 |
| | OOD | 13.0 | 15.0 | 16.0 |
| 0.40 | ID | 5.0 | 11.0 | 15.0 |
| | OOD | 5.0 | 5.0 | 11.0 |

specifically developed for such workflows, where reliability and privacy are prioritised over speed, and these latencies remain practically acceptable. To provide a complete view of computational cost, we report both communication rounds (Appendix G) and wall-clock runtimes for comparison with other methods. Together, these results help practitioners assess whether DIsoN is suitable for their deployment setting.

DIsoN introduces additional computation at inference time (approximately 40 s–4 min per sample in our experiments on an NVIDIA RTX A5000), as each target sample requires iterative isolation training until convergence. In contrast, most compared baselines complete inference within milliseconds, since they rely on a single forward pass of a CNN backbone followed by lightweight post-processing to compute an OOD score (e.g., distance computation, linear projection, or tree traversal). Note that iForest, conceptually the closest to DIsoN, cannot be trained jointly on source and target data due to privacy constraints, unlike DIsoN. Following the original iForest formulation [25], we pre-train it only on the source (ID) data and then apply it with a single forward pass per target sample to determine whether it is ID or OOD, which explains its much lower runtime.

While this makes the baselines faster, DIsoN's iterative optimisation results in more reliable separation between ID and OOD samples, as demonstrated in our results. Average per-sample runtimes for representative baselines on the Ultrasound dataset are reported in Table 12.

Table 12: Average inference time per target sample (seconds, NVIDIA RTX A5000). Baseline methods compute OOD scores after a single forward pass of the pre-trained network followed by lightweight post-processing, whereas CC-DIsoN performs iterative optimisation until convergence, which increases latency but yields more reliable OOD detection performance.

| Method | Avg. Runtime (s) |
|---|---|
| MDS | 0.006 |
| ViM | 0.005 |
| iForest | 0.015 |
| CIDER | 0.007 |
| PALM | 0.007 |
| CC-DIsoN (ID samples) | 155.2 |
| CC-DIsoN (OOD samples) | 90.9 |

**Single-Sample vs. Batch Processing.** DIsoN is designed to process one target sample at a time, as it learns a decision boundary that separates a single target sample from the training distribution using convergence behaviour as the OOD score. This design aligns with many real-world scenarios, especially in healthcare, where each patient scan is acquired and analysed individually rather than in large batches. If batch processing is feasible, multiple DIsoN instances can be executed in parallel to use available hardware efficiently. In our experiments, up to eight DIsoN runs could be executed concurrently on a single NVIDIA RTX A5000 (24 GB VRAM). Although not a traditional deep learning batch-tensor setup, this parallel execution allows scaling for pre-collected datasets.

**Network Conditions and Communication Latency.** DIsoN involves a single target client communicating with a source node, making it simpler and more robust than large-scale multi-client federated systems. Network delays or volatility only affect wall-clock time, not the number of communication

rounds or the quality of convergence. In practice, network disconnections can be handled through standard retry or timeout mechanisms used in distributed systems.

Overall, these experiments and considerations demonstrate that while DIsoN trades inference speed for reliability, its runtime remains predictable and manageable across deployment conditions, making it suitable for safety-critical settings where performance is prioritised and real-time inference is not required.

# I    Additional Multi-Seed Results for Histopathology

Due to space limitations, Table 2 reports means over three seeds only, the standard deviations are omitted. In this section, we provide the complete tables including standard deviations. Tables 13 (AUROC $\uparrow$) and 14 (FPR95 $\downarrow$) list the *mean $\pm$ std* over three random seeds for the Histopathology dataset. For the Avg. columns, the $\pm$ values are computed as the *sample standard deviation across seeds of the per-seed macro averages*.

Table 13: Complete AUROC ($\uparrow$) results for the Histopathology dataset (MIDOG). Each cell shows mean $\pm$ standard deviation over three random seeds. This extends the means reported in Table 2 with standard deviation.

| Methods | near-OOD | | | | | | | | far-OOD | | |
|---|---|---|---|---|---|---|---|---|---|---|---|
| | 2 | 3 | 4 | 5 | 6a | 6b | 7 | Avg. | CCAgT | FNAC | Avg. |
| MSP | $54.1_{\pm5.8}$ | $47.4_{\pm13.0}$ | $55.7_{\pm2.5}$ | $59.1_{\pm1.2}$ | $54.8_{\pm0.6}$ | $45.6_{\pm9.3}$ | $56.1_{\pm5.3}$ | $53.3_{\pm3.5}$ | $78.4_{\pm1.4}$ | $82.7_{\pm4.3}$ | $80.6_{\pm1.6}$ |
| MDS | $67.4_{\pm3.0}$ | $67.2_{\pm0.6}$ | $65.7_{\pm0.3}$ | $61.2_{\pm4.5}$ | $60.6_{\pm1.6}$ | $55.6_{\pm5.9}$ | $51.0_{\pm7.1}$ | $61.2_{\pm1.8}$ | $87.1_{\pm9.9}$ | $86.6_{\pm11.9}$ | $86.8_{\pm10.5}$ |
| fDBD | $57.9_{\pm8.2}$ | $52.6_{\pm8.4}$ | $57.5_{\pm4.3}$ | $60.2_{\pm2.4}$ | $57.9_{\pm1.6}$ | $51.2_{\pm6.1}$ | $55.3_{\pm4.5}$ | $56.1_{\pm3.8}$ | $74.6_{\pm13.8}$ | $82.1_{\pm0.6}$ | $78.3_{\pm7.1}$ |
| ViM | $68.0_{\pm3.6}$ | $66.7_{\pm2.6}$ | $66.2_{\pm1.2}$ | $63.5_{\pm1.4}$ | $61.9_{\pm0.9}$ | $55.7_{\pm7.0}$ | $54.9_{\pm5.5}$ | $62.4_{\pm1.5}$ | $92.5_{\pm3.4}$ | $92.6_{\pm5.2}$ | $92.6_{\pm4.0}$ |
| iForest | $37.9_{\pm5.8}$ | $38.7_{\pm4.8}$ | $39.7_{\pm1.9}$ | $41.6_{\pm6.5}$ | $39.5_{\pm1.9}$ | $40.4_{\pm4.3}$ | $46.4_{\pm9.4}$ | $40.6_{\pm1.9}$ | $27.7_{\pm16.4}$ | $32.3_{\pm24.3}$ | $30.0_{\pm20.2}$ |
| CIDER | $71.4_{\pm3.3}$ | $65.6_{\pm8.1}$ | $55.5_{\pm6.5}$ | $64.2_{\pm2.5}$ | $57.4_{\pm2.0}$ | $47.7_{\pm10.0}$ | $64.1_{\pm3.0}$ | $60.9_{\pm2.2}$ | $82.5_{\pm4.4}$ | $95.4_{\pm3.4}$ | $89.0_{\pm2.5}$ |
| PALM | $73.5_{\pm2.3}$ | $59.2_{\pm11.0}$ | $66.3_{\pm5.5}$ | $64.3_{\pm1.1}$ | $62.1_{\pm3.9}$ | $41.8_{\pm4.5}$ | $61.6_{\pm2.5}$ | $61.3_{\pm3.4}$ | $97.0_{\pm4.0}$ | $99.6_{\pm0.6}$ | $98.3_{\pm2.3}$ |
| CC-DIsoN | $75.4_{\pm1.3}$ | $79.5_{\pm1.6}$ | $72.6_{\pm0.6}$ | $63.0_{\pm1.7}$ | $64.0_{\pm2.3}$ | $70.7_{\pm2.8}$ | $61.8_{\pm2.0}$ | $69.6_{\pm0.7}$ | $98.3_{\pm0.1}$ | $98.4_{\pm0.1}$ | $98.3_{\pm0.1}$ |

Table 14: Complete FPR95 ($\downarrow$) results for the Histopathology dataset (MIDOG). Each cell shows mean $\pm$ standard deviation over three random seeds. This extends the means reported in Table 2 with standard deviation.

| Methods | near-OOD | | | | | | | | far-OOD | | |
|---|---|---|---|---|---|---|---|---|---|---|---|
| | 2 | 3 | 4 | 5 | 6a | 6b | 7 | Avg. | CCAgT | FNAC | Avg. |
| MSP | $94.8_{\pm4.2}$ | $93.4_{\pm7.0}$ | $89.4_{\pm5.4}$ | $93.5_{\pm0.6}$ | $94.7_{\pm1.4}$ | $95.5_{\pm4.4}$ | $92.4_{\pm3.5}$ | $93.4_{\pm1.9}$ | $52.2_{\pm10.0}$ | $63.6_{\pm8.4}$ | $57.9_{\pm5.4}$ |
| MDS | $80.0_{\pm3.0}$ | $79.8_{\pm1.6}$ | $81.9_{\pm5.0}$ | $87.4_{\pm3.4}$ | $87.4_{\pm2.8}$ | $89.5_{\pm2.4}$ | $91.6_{\pm2.9}$ | $85.4_{\pm2.7}$ | $43.6_{\pm20.6}$ | $39.6_{\pm26.6}$ | $41.6_{\pm23.1}$ |
| fDBD | $83.0_{\pm7.2}$ | $81.4_{\pm8.8}$ | $88.7_{\pm2.6}$ | $89.0_{\pm2.7}$ | $85.9_{\pm7.9}$ | $86.2_{\pm9.8}$ | $92.2_{\pm1.3}$ | $86.6_{\pm4.8}$ | $58.3_{\pm10.8}$ | $63.7_{\pm2.9}$ | $61.0_{\pm6.8}$ |
| ViM | $79.8_{\pm7.3}$ | $77.8_{\pm4.3}$ | $77.7_{\pm1.8}$ | $86.9_{\pm3.1}$ | $87.3_{\pm0.7}$ | $91.0_{\pm3.4}$ | $93.1_{\pm2.8}$ | $84.8_{\pm2.6}$ | $29.1_{\pm9.8}$ | $27.0_{\pm16.2}$ | $28.0_{\pm12.6}$ |
| iForest | $98.3_{\pm2.0}$ | $96.9_{\pm2.4}$ | $97.7_{\pm1.8}$ | $97.5_{\pm1.2}$ | $98.0_{\pm1.7}$ | $97.7_{\pm2.9}$ | $95.7_{\pm1.2}$ | $97.4_{\pm1.4}$ | $99.2_{\pm0.4}$ | $96.8_{\pm4.1}$ | $98.0_{\pm2.1}$ |
| CIDER | $84.6_{\pm4.7}$ | $89.0_{\pm3.8}$ | $91.5_{\pm1.9}$ | $89.8_{\pm2.2}$ | $92.2_{\pm3.0}$ | $96.5_{\pm0.5}$ | $88.2_{\pm5.1}$ | $90.2_{\pm2.4}$ | $77.2_{\pm16.5}$ | $18.3_{\pm13.3}$ | $47.8_{\pm6.2}$ |
| PALM | $78.1_{\pm5.1}$ | $90.3_{\pm1.8}$ | $77.7_{\pm7.4}$ | $93.6_{\pm0.4}$ | $90.8_{\pm1.2}$ | $97.6_{\pm1.4}$ | $93.6_{\pm3.1}$ | $88.8_{\pm0.8}$ | $21.8_{\pm35.7}$ | $1.5_{\pm2.5}$ | $11.6_{\pm19.1}$ |
| CC-DIsoN | $78.8_{\pm4.0}$ | $61.7_{\pm3.6}$ | $79.7_{\pm2.7}$ | $89.4_{\pm2.4}$ | $85.3_{\pm5.2}$ | $79.9_{\pm5.0}$ | $92.0_{\pm0.0}$ | $81.0_{\pm1.2}$ | $4.8_{\pm1.8}$ | $4.0_{\pm1.4}$ | $4.4_{\pm1.6}$ |

# J    Qualitative Visualization of DIsoN Isolation Process

In this section, we provide qualitative visualizations of the DIsoN isolation process over communication rounds. Figure 7 shows PCA projections of a target sample and the source data after communication rounds 0, 15, and 25. Each row corresponds to a target image from the Dermatology dataset. In the OOD example (top row), the target sample (orange star) rapidly drifts away from the source distribution (blue points), achieving clear separation already after communication round 15. In contrast, the ID target (bottom row) remains closely clustered with the source data, showing that it is harder to isolate. This side-by-side comparison shows DIsoN's core idea/motivation: OOD samples isolate quickly, while ID samples require more rounds.

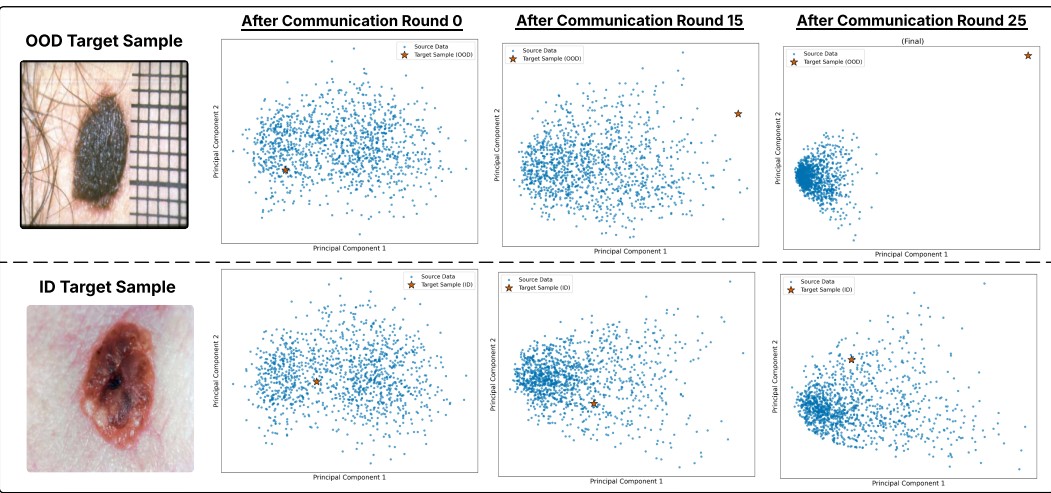

Figure 7: **Isolation Process over Communication Rounds.** PCA projections of source samples (blue) and a target sample (orange star) after communication rounds 0, 15, 25 of DIsoN training. The top row shows a target sample that is OOD, the bottom row shows a target sample that is ID (from the Dermatology dataset). The OOD sample becomes separated already after round 15 and is clearly isolated by round 25. In contrast, the ID sample remains entangled with the source distribution throughout, demonstrating that it is harder to isolate.

