# OpenReview forum: "DIsoN: Decentralized Isolation Networks for Out-of-Distribution Detection in Medical Imaging"
_NeurIPS.cc/2025/Conference — NeurIPS 2025 poster_

### Official Review · Reviewer_8S2S · 2025-06-22

**Clarity:** 2
**Significance:** 3
**Originality:** 3
**Rating:** 5
**Confidence:** 4

**Summary:**

The paper proposes a non-parametric out-of-distribution (OOD) detection method that scores test samples based on the convergence time of a binary classifier neural network trained to distinguish the test example from the training dataset. The framework is built under decentralized assumptions and can be deployed in private environment, extinguishing data sharing and only allowing for parameter sharing between server and client.

**Questions:**

1. What if the predicted class is wrong? Could the authors elaborate how much it impacts CC-DIsoN? Showcasing robustness under misclassification could improve the method's trustworthiness.
2. Which other methods listed in Table 1 employ a class-conditional criterion? Clarifying this point would help readers better understand how DIsoN compares to existing approaches.
3. Why were $E_{stab} = 5$ and $\tau = 0.85$ chosen? Providing more detail on the rationale behind these parameter choices would strengthen the methodology section.
4. Could the authors please elaborate on why the isolation forest exhibits such low performance, with AUROC values even below 0.5 in Table 2? Although you cite isolation forests as inspiration for isolation networks, the results for iForest are poor. Some discussion or analysis regarding this discrepancy would be helpful.
5. In Figure 4, only the average number of rounds to convergence is reported. Could you provide the full distribution of this metric? Reporting the median, as well as other quantile statistics, could be more informative, as there may be significant variability and some samples might be much harder to isolate than others. Presenting this additional information would help practitioners better assess whether the method is suitable for their specific applications.

I'm willing to increase my score if these concerns are satisfactorily addressed.

**Ethical Concerns:**

["NO or VERY MINOR ethics concerns only"]

**Final Justification:**

The authors have answered all points I raised during the rebuttal. I've read the other reviews and I don't see strong reasons to reject. Thus, I increase my score towards acceptance.

**Limitations:**

Authors briefly mention limitations of their work. The proposed method is costly and lacks adequate runtime analysis.

**Quality:**

3

**Strengths And Weaknesses:**

### Strengths:
The paper addresses an important limitation of non-parametric methods for OOD detection: data privacy concerns. It also introduces a novel sample-wise optimization-based approach for OOD detection. The proposed method demonstrates strong performance on a relevant medical image classification benchmark.

### Weaknesses:
- Limited mathematical formulation. In Section 3, the preliminary formulation does not include labels or pseudo-labels, even though the method relies on class-conditional information. Providing a more comprehensive formulation would improve clarity.
- Insufficient ablation study on class-conditional importance. There is a limited discussion of how class-conditional information contributes to the method's effectiveness.
- Limited discussion of algorithm runtime. The paper only briefly mentions runtime in the conclusion. A more thorough evaluation of runtime would help practitioners determine the method's applicability and would strengthen the paper's contributions.
- Minor: writing. The writing could be improved. Excessive use of passive voice makes some passages difficult to read.

---

> ### Author Rebuttal · Authors · 2025-07-31
>
> Dear reviewer, thank you for your effort reviewing this work and the
> constructive feedback! We are glad that you found our work
> addresses an "**important limitation of non-parametric methods for OOD
> detection: data privacy concerns**", acknowledging its originality as it
> "introduces a **novel**" method for OOD detection with "**strong
> performance**".
> We address your points below, and added these clarifications and results to the revised paper (using the extra camera-ready page).
>
> ## [W1] Mathematical preliminary formulation does not include labels?
>
> Apologies, we’re not 100% sure what caused this confusion. The
> preliminary math notation defines labels space $\mathcal{Y}$ and
> (true) labels $y_i$ early on (Sec.3 L133-5), with predicted class (pseudo-label) defined later in Sec. 3.3 L242. We do not define pseudo-labels earlier since they don't affect the derivation of intermediate theory. We agree clarity can be improved though and added notation for pseudo-labels at the start of Sec.3, disambiguated notation of true and pseudo-labels and ID/OOD prediction, and added explanations in Sec.3.3.
>
> ## [W2] Limited ablation study / discussion about class-conditional sampling
> ## And
> ## [Q1] What if the predicted class is wrong?
>
> CC-DIsoN uses predictions of a primary model $M_{pre}$ deployed
> for a task of interest (e.g. classify cardiomegaly in Xrays, Sec.4) as
> pseudo-labels for class-conditional sampling. This allows comparing the target sample to the closest relevant cluster (predicted class) rather than all training data. The original ablation (Tab. 3) showed class-conditional sampling helps, and DIsoN without CC-sampling
> still works well (compare Tab. 3 with Tab. 1 baselines).
>
> Your suggestion to explore misclassification effects is insightful and led to the following experiments.
>
> To assess misclassification rates, we report the pre-trained primary models $M_{pre}$'s accuracy on target ID, OOD, and combined samples (col. 1-3). To deploy a model in real-world deployment a high ID accuracy is required. Generalisation is impeded by domain-shift, however, thus predictions for OOD samples are expected less accurate than on ID. We compare for trends against performance of CC-DIsoN for OOD detection (col.4).
>
> |Dataset|Classif Acc. on ID target samples|Classif Acc. on OOD target samples|Classif Acc. on ID+OOD target samples|Performance on OOD detection by CC-DIsoN (AUROC, Tab.1)|
> |-|-|-|-|-|
> |Dermatology|77.6%|54.0%|63.0%|87.96|
> |X-Ray|93.0%|70.8%|82.2%|83.94|
> |Ultrasound|78.6%|61.4%|64.8%|66.58|
> |MIDOG (near+far)|77.3%|68.6%|69.2%|75.50|
>
> CC-DIsoN's performance does not correlate with classification acc.. E.g. Dermatology shows highest AUROC (87.96) despite lowest classifier acc. (63), suggesting
> that performance depends more on data type than classifier acc. .
>
> To further investigate misclassification effects, we designed
> 3 experiments manually assigning wrong class pseudo-labels for certain samples:
>
> - **Predict wrong class for all data**: We assign wrong class as pseudo-labels for \*all\* target samples, regardless if ID or OOD. This represents the extreme case where the primary classifier has 0% acc. . Such model would never be deployed in real-world.
>
> - **Predict wrong class for ID data**: All target ID samples are assigned wrong class, while OOD samples use predicted classes. Central motivation for class-conditioned sampling was to make isolation more difficult for ID target samples, as they should be harder to isolate from source (training ID) samples of the same class, due to common visual features. This should lead to slower convergence and increase the difference than when separating OOD samples from source (training ID) samples, even of the same class, which is faster due to domain-shift. If we instead compare ID target samples to source (training) samples from the wrong class, the isolation will be easier, resulting in worse OOD detection.
>
> - **Predict wrong class for OOD data**: All target OOD samples are assigned a wrong class, while ID samples use the predicted class. Domain-shift degrades acc. of ML models on OOD data. Hence this experiment simulates the most extreme cases of domain shift. Since OOD samples already differ visually from the source distribution, comparing them to unrelated classes will make their convergence even faster, thus we expect easier OOD detection.
>
> |Dataset|Model|AUROC (%) ↑|
> |-|-|-|
> |**Dermatology**|Best baseline|83.4|
> ||DIsoN|86.4|
> ||CC-DIsoN|88.0|
> ||ID+OOD wrong|83.1|
> ||ID wrong|84.7|
> ||OOD wrong|86.7|
> |**Breast Ultrasound**|Best baseline|61.0|
> ||DIsoN|67.0|
> ||CC-DIsoN|66.6|
> ||ID+OOD wrong|61.5|
> ||ID wrong|34.6|
> ||OOD wrong|89.6|
> |**Chest X-Ray**|Best baseline|77.7|
> ||DIsoN|81.4|
> ||CC-DIsoN|83.9|
> ||ID+OOD wrong|77.9|
> ||ID wrong|74.0|
> ||OOD wrong|88.3|
>
> The table confirms expected results: CC-DIsoN's OOD detection AUROC degrades when ID samples are mislabeled, and tends to increase when OOD samples are. Even under extreme and impractical cases of misclassification, CC-DIsoN outperforms the best baseline (Tab. 1 in paper) in all but one case, demonstrating robustness. In practice, especially in safety critical domains like healthcare, we would only deploy methods with high ID acc. But domain-shift of OOD samples is always an issue for performance. Hence CC-DIsoN's design and behavior is appropriate. Notably, DIsoN (without CC sampling) also outperforms the best
> baseline.
>
> We added this ablation to Sec.4.1,
> extended discussion on CC sampling, and added classification accuracies
> (top table) to Appendix. Many thanks, this strengthens the analysis of CC sampling a lot.
>
> ## [W3] Limited discussion of algorithm runtime
>
> ## and
>
> ## [Q5] More info about number of rounds to convergence (medians/quantiles)
>
> We agree that more details could help practitioners. As requested, we provide below medians and quantiles of the distribution of convergence rounds (for default $\alpha=0.8$).
> Since wall-clock runtime is hardware-dependent, we report number of rounds. Reporting ID and OOD separately helps practitioners estimate runtime based on their expected ID/OOD ratio.
>
> For the camera-ready, we will extend this to other $\alpha$ values and visualise them in Fig.4, with the full table in the Appendix.
>
> |Dataset|Sample Type|25th Perc.|Median|75th Perc.|
> |-|-|-|-|-|
> |Dermatology|ID|68.0|126.5|184.0|
> ||OOD|22.0|28.0|40.0|
> |Ultrasound|ID|20.0|29.0|100.0|
> ||OOD|16.0|20.5|32.0|
> |X-Ray|ID|28.0|41.0|92.0|
> ||OOD|16.0|20.0|26.0|
> |Histop. (near)|ID|44.3|59.5|74.0|
> ||OOD|34.0|44.0|57.0|
>
> ## \[W4\] (Minor) Improve writing / passive voice
>
> Thank you for the advice, we revised phrasings and reduce passive
> voice and improve flow where helpful.
>
> ## Questions:
>
> ## Q1 and Q5 were answered above.
> \
>
> ## Q2: Which other methods in Table 1 employ a class-conditional criterion?
>
> To avoid any possible misunderstanding, we clarify that our method \*never\* uses any real labels of the \*true\*
> class for the target samples to perform Class-Conditional sampling (or otherwise). It only uses the predictions of the primary model $M_{pre}$ as pseudo-labels. Hence, it has no advantage over other methods in terms of input information for the target samples.
>
> Using its class predictions as part of OOD detection method is common in
> literature. Methods differ in how explicitly or implicitly they do it.
>
> Among the compared methods, MDS measures Mahalanobis distance of a
> target sample from the cluster of the closest class (=model's predicted class), thus implicitly using a "class-conditioning
> criterion". PALM and CIDER also use Mahalanobis distance. MSP uses the predicted class's softmax probability. fDBD measures distance to the decision boundary
> thus also implicitly the model's class prediction. ViM explicitly uses class-information, via the max logit of the predicted class. Only iForest does not use class predictions, which is a limitation. Since pseudo-labels are model-derived (no ground-truth), our method has no advantage over the compared methods. We agree this is worth clarifying and added this clarification in Sec.4.
>
> ## Q3: How are values $E_{stab}=5$ and $tau=0.85$ chosen?
>
> Intuitively, these parameters capture the convergence of the isolation
> process. $E_{stab}$ acts as a "patience" parameter,
> conceptually similar to patience in learning-rate schedulers, requiring
> correct classification for 5 consecutive rounds. Value 5 is common in
> schedulers to capture convergence. With $\tau$ we require the model to have learned to separate the target sample confidently enough, giving softmax probability \>0.85. The
> value 0.85 was chosen because it's common in uncertainty-based
> literature (e.g.\[3\]). In preliminary experiments these values worked well enough, thus
> we kept them constant and did not calibrate them further to avoid overfitting to our data. We added explanations for the above to Sec. 3.1.
>
> ## Q4: Isolation Forest does not perform well?
>
> Isolation Forests were originally developed for tabular data, were they
> were found to perform well and built their reputation. For imaging data,
> however, they are often found to not perform well (e.g. \[1,2\]),
> therefore this is no surprise in our study. We are inspired by ideas,
> however, not performance numbers. Hence one of the motivations of our
> work was to develop an approach that exploits the great idea of
> 'isolation' in iForests and makes it work well for imaging with
> networks. We added a clarification about iForest performance in Sec.
> 4.1.
>
> Thank you again for the detailed points to improve the paper! We believe
> the additional experiments and clarifications address your concerns.
> Please let us know if any questions remain and we are happy to engage in
> further discussion.
>
> ## References
>
> [1] Sun et al, Out-of-distribution detection with deep nearest
> neighbors, 2022
>
> [2] Fan et al, Test-Time Linear Out-of-Distribution Detection, 2024
>
> [3] Kamnitsas et al, Transductive image segmentation: Self-training
> and effect of uncertainty estimation, 2021

---

> > ### Comment · Reviewer_8S2S · 2025-08-05
> >
> > Dear authors.
> > Thank you for this detailed and thorough rebuttal. I specially appreciate the analysis on the class conditional detection.
> > You have answered all points I raised and I'll raise my score accordingly.

---

> > > ### Author Response · Authors · 2025-08-06
> > >
> > > Dear reviewer,
> > >
> > > thank you for your positive response! We are glad to hear we addressed all your points, especially with the added investigation of class-conditional sampling. Many thanks again for your valuable feedback, the additional analysis and clarifications have strengthened the paper.

---

> ### Comment · Area_Chair_TJUy · 2025-08-05
> **Feedback Needed - Your AC**
>
> Dear Reviewer 8S2S,
>
> I notice that the authors have submitted a rebuttal. Could you please let me know if the rebuttal addresses your concerns? Your engagement is crucial to this work.
>
> Thanks for your contribution to our community.
>
> Your AC

---

### Official Review · Reviewer_vx6Z · 2025-06-26

**Clarity:** 2
**Significance:** 3
**Originality:** 2
**Rating:** 4
**Confidence:** 4

**Summary:**

This paper presents DIsoN, a decentralized solution for out-of-distribution (OOD) sample detection in medical imaging. It leverages an Isolation Network to assess OOD samples based on convergence speed and enables collaborative training between source and target nodes without sharing raw data. Experimental results demonstrate that DIsoN achieves significant improvements in OOD detection performance. However, the article has some shortcomings in terms of expression clarity, selection of experimental datasets, and comparison methods.

**Questions:**

See Weaknesses
1.	Improving the clarity of the article's expression will make it easier for the audience to understand.
2. The authors should include a discussion on few-shot federated learning. It is strongly recommended to incorporate a comparison with recent few-shot federated learning methods， such as [1]. This will provide a more comprehensive assessment of the effectiveness of your method.
[1] FedAffect: Few-Shot Federated Learning for Facial Expression Recognition, ICCV Workshops, 2021.
3. While the authors discuss the impact of /alpha on performance and speed in the ablation study, the values of |Bs| and N differ significantly across datasets. Could the authors provide more details? Clarifying the amount of data required for fine-tuning is crucial for the practical application of the method.
4. Why did the authors choose generated data for OOD detection? Is it possible to use similar real-world datasets as OOD data to demonstrate the method's effectiveness on actual images, including natural images?

**Ethical Concerns:**

["NO or VERY MINOR ethics concerns only"]

**Final Justification:**

Most of the concerns have been addressed. I will thus increase my score to weak accept.

**Limitations:**

See Questions.

**Quality:**

3

**Strengths And Weaknesses:**

Strengths:
1. The overall structure of the article is well-organized, and the images in this article effectively illustrate the method's flow.
2. The proposed method utilizes convergence rate as an indicator for OOD detection and incorporates collaborative learning to achieve online inference while ensuring privacy protection.

Weaknesses:
1.	There are some typos, including a misspelling of "odd" on line 45. Additionally, the sentences on lines 155-157 lack clarity and ambiguity, particularly regarding the data composition of SN and TN.
2.	While the paper highlights its distinctions from federated learning, the proposed setup resembles a two-client federated learning scenario, with aggregation occurring on a few-shot client. The derivation of formulas also represents a special case of FedAvg. The authors should discuss and compare their work with few-shot federated learning.
3.	The datasets (Dermatology, Breast Ultrasound, and Chest X-Ray) used artificial artifacts as OOD data, resulting in significant differences. While Table 1 demonstrates the method's effectiveness, Table 2 shows that the advantages of the proposed method are not as pronounced with real medical images. The authors should analyze the reasons for this discrepancy and consider using real medical image data from different domains for OOD tasks to construct a more robust test dataset.
4.	OOD images often exhibit strong similarities to source domain images, making fine-tuning the backbone prone to overfitting. An interesting practical approach is to fine-tune the classification head instead. The authors could enhance the discussion around this practice to underscore the practical significance of the method.

---

> ### Author Rebuttal · Authors · 2025-07-31
>
> Dear reviewer, thank you for your review and feedback. We are glad that
> you acknowledge our work is "well organized" and it achieves
> "**significant improvements**" in OOD detection. With all due respect,
> we believe the reviewer has misunderstood important aspects of the work,
> especially regarding the data for evaluation (misunderstood as
> artificial) and method performance (characterized "not as pronounced" in
> Table 2), which did not allow the reviewer to appreciate the work's
> significance. We address these points below, and sincerely urge the
> reviewer to reconsider the novelty of the OOD framework, strength of
> evaluation and overall significance of the work and assigned reviewer
> score.
>
> ## [W1] and [Q1]: Typos & Clarity of L155-157 regarding data composition in SN & TN
>
> Thank you for raising this. We will carefully proofread and fix typos and unclear phrasing.
>
> We appreciate you pointed out that L155-157 confused you regarding data
> composition on Source Node (SN) and Target Node (TN). We want to clarify, SN is defined in Sec.3.2, L177 and TN is defined in Sec.3.2, L180. Sec 3.1 and L155-157 do not relate to SN and TN, which are introduced only later for the decentralized setting in Sec.3.2. Instead, because the Isolation Network algorithm is entirely novel, we first derive its
> centralized version in Sec.3.1, before describing the more complex decentralized version (Sec.3.2).
>
> Thus L155–157 describe the composition of a training batch $B$ in a centralized setting (stated in L150). To disambiguate L155–157 as requested, we introduce math notation for the batch to complement the natural-language description. Specifically, a batch $B= B_s \cup ( x_t^{(1)}, ..., x_t^{(N)} )$, where $B_s$ is a set of $|B_s|$ source samples from $D_s$, and all $x_t^{(i)}$ are $N$ copies of the same target sample $x_t$ (oversampling it to counter ID/OOD imbalance in training batches and avoid trivial optimization solutions, cf. L154). We will also clarify in Sec. 3.1 that this refers to a centralized setting and make its connection to the decentralized SN/TN setup clearer.
>
> ## [W3] & [Q3]: Misunderstanding that Datasets used are synthetic and inappropriate
>
> The reviewer states as main weakness (W2) "The datasets (Dermatology,
> Breast Ultrasound, and Chest X-Ray) used artificial artifacts as OOD
> data" and "consider using real medical image data from different domains
> for OOD tasks to construct a more robust test dataset.". They ask why we
> used "[generated data]" (Q3). Apologies but we believe this is a
> major misunderstanding.
>
> All data we used, **including** X-Rays, Breast Ultrasound, Dermatology,
> are real-world clinical data, collected from real clinics, and widely
> used (see references to each clinical dataset in Sec. 4). All these OOD
> artifacts are naturally present in the original data. No external
> modifications or synthetic patterns were added/used. Although the
> reviewer may be more accustomed to different types of domain shifts
> (e.g. whole image quality/appearance change due to different
> camera/scanner), data with localised OOD artifacts as used herein (e.g.
> pacemakers in Xrays) are often employed as OOD sets in literature, eg.
> [3]).
>
> We believe this misunderstanding is from misinterpreting [2] and [3] that we cite as sources of the datasets. [2,3] separated which images in real clinical databases (e.g. CheXPert) have abnormal artifacts and which do not, to create OOD and ID subsets, which we use in this study. They [2,3] also created some *synthetic* data to perform their OOD analysis but we do not use them. We only use real ID and OOD images (see Sec.4, real skin rulers, ultrasound annotations, Xray pacemakers). We updated Sec.4 to state that we only use the real data.
>
> We do not include Natural images that reviewer suggests, as it is outside our research agenda. We find that clinical data, with their unique challenges, are fantastic benchmarks and inspire innovation (novelty and performance acknowledged by other reviewers).
>
> ## [W3 & Q3] Unconvincing performance on real medical data
>
> As per the above, the reviewer misunderstood that Table 1 is artificial data, hence not convincing.
> This is coupled by what we think is a second misunderstanding, when the reviewer states (W3) that Table 2 “shows that the advantages of the proposed method are not as pronounced”.
>
> We point out that in Table 2, on average (Column “Avg”) we achieve improvement of 5.5% AUROC compared to the best SOTA method and lowest FPR95 for near-OOD artifacts.
> In far-OOD we surpass 6 out of 7 methods, and only PALM surpasses our method by 1% AUROC.
>
> Perhaps the reviewer got confused when reading performance on Table 2,
> because FPR is best when it is low, but AUROC is best when high? We
> noted this as up/down arrows in captions, but will state it explicitly
> in updated manuscript.
>
> We hope this clarification shows our method's
> very strong performance on real medical data, and we urge the reviewer to
> reconsider their overall opinion about our method, which may have been
> affected/disappointed due to the above misunderstandings.
>
> ## [W2, Q2] Discuss/compare with Few-Shot Federated Learning
>
> The reviewer raises a similarity with Few-Shot Federated Learning. While we agree DIsoN’s decentralized training shares similarities with 2-client Federated Learning (FL) (acknowledged in L123, “DIsoN uses decentralised optimization similar to Federated Learning”), there are fundamental differences. Objective, data composition, and output fundamentally differ from FL and Few Shot FL:
>
> **Optimization objective:**
>
> a. **Few-Shot FL:** Trains a model for a specific task (e.g. disease classification) with the aim of **generalizing (predicting)** new unseen data.
>
> b. **DIsoN for OOD:** Generalization is not a goal. We optimize a model to separate 1 target sample from the source data to determine if it is OOD.
>
> **Data Composition:**
>
> a. Few-Shot FL assumes clients have limited labels for different classes (and possibly unlabeled data, eg. FedAffect [REF1]).
>
> b. In DIsoN the Source Node has a big dataset, with all samples assigned label 0, indicating it is the *source*. The Target Node has only 1 sample, labelled 1 to indicate the *target*. Client has no labels of different classes.
>
> **Output:**
>
> a. **Few-Shot FL:** Produces a generalizable model to make predictions on new data.
>
> b. **DIsoN:** Trained model is discarded. Only rounds to convergence is useful as OOD score.
>
> Few-Shot FL methods (including FedAffect [REF1] provided by reviewer) are not designed for this setting and OOD detection. Thus, comparison is not possible. We will add explanation about the relation with Few-Shot FL to Sec.2 to clarify for readers.
>
> ## [W4] Fine-Tuning only Classification Head to avoid over-fitting.
>
> Freezing the backbone assumes that the pre-trained feature extractor
> has sufficiently expressive embeddings to distinguish ID from OOD
> samples. This may not be the case as the isolation task of a single
> target sample is distinct from common tasks where the model is trained
> to fit large data. Thus allowing optimization of all embeddings may be
> beneficial. In fact, in early (not reported) experiments in Dermatology
> we explored freezing parts of the network:
>
> - Head-only finetuning: Unstable, not appropriate convergence.
>
> - Head-Only + Final ResNet Block finetuning: AUROC: 84.23% / FPR95: 65.49%
>
> - Full Network finetuning (our method): AUROC: 87.96% / FPR95: 40.14% (both metrics better)
>
> Finetuning only the head did not work, while finetuning head and final
> ResNet block was worse than fine-tuning whole model, confirming the
> above intuition. So we did not pursue those further.
>
> We **added a discussion about fine-tuning** the classification head to
> our paper to clarify this.
>
> Finally, we note that our work has introduced a novel
> (decentralised) OOD framework that trains an isolation network for
> each target sample separately, and uses convergence speed as OOD
> score, not to make generalizable predictions on new data. Overfitting and regularization play a different role in this setting. We already explored one
> regularization method, stochastic feature augmentations (Sec.3.2, and experiments
> in Sec.4.2). While further exploration is interesting, the possible methods are
> endless, and there is no strong evidence that partial finetuning is the
> next best step. We urge the reviewer to consider that the work has introduced a novel algorithm with promising performance and already considered analysis of a
> regularization method. Thus this should not be seen as a weakness, but
> avenues for future work inspired by the intriguing nature of this novel
> algorithm, which warrants its publication.
>
> ## [Q3] Details why values of \|Bs\| and N differ across datasets?
>
> Allow us to clarify, as this may also be a small misunderstanding. $|B_s|$ refers to the number of source samples per mini-batch used at SN and is fixed to 16 for all datasets except Breast Ultrasound, where it is 8 due to smaller dataset size (see Appendix, Sec. D – Table 6).
> N denotes how often a target sample is repeated (L157) in an idealized centralized setting (see response to your [W1, Q1] as the misunderstandings may be related). N is not explicitly used in our experiments, because we only use the final decentralized DIsoN algorithm (not centralized), where N is not used.
> Instead, in the decentralised algorithm, N is implicit as a function of $\alpha$. This is shown by Proposition 3.1 with $\alpha = \frac{|Bs|}{(|Bs|+N)}$, where defining value of $\alpha$ implicitly defines N. Given that we use a fixed $\alpha =0.8$ across all datasets (see Fig 4.a for choice of $\alpha$), and $|B_s|=16$ except for Ultrasound with $|B_s|=8$, we can derive N=4 for all datasets except Ultrasound N=2. Therefore, N also does not vary except for the smaller ultrasound dataset.
>
> **References**
>
> **[REF1]** FedAffect: Few-Shot Federated Learning for Facial
> Expression Recognition, ICCV Workshops, 2021

---

> > ### Comment · Reviewer_vx6Z · 2025-08-04
> >
> > Thank you for the detailed response. Most of my concerns have been addressed. I will thus increase my score.
> > Please involve the discussions in the revision.

---

> > > ### Author Response · Authors · 2025-08-04
> > >
> > > Thank you very much. We are pleased to know this has addressed your concerns and clarified any potential misunderstandings.
> > > We absolutely commit to updating the manuscript for the camera-ready version with the clarifications and information above, as also stated in our response, as we believe these additions will be useful to enhance clarity for future readers and support their understanding of our work.
> > >
> > > Thank you again,
> > >
> > > The authors

---

### Official Review · Reviewer_SHNi · 2025-06-26

**Clarity:** 2
**Significance:** 3
**Originality:** 3
**Rating:** 4
**Confidence:** 3

**Summary:**

The paper introduces an original OOD detection method, Isolation Network. The core proposition is that it would be beneficial to train a binary classifier to provide an estimation of how difficult, for example, it would be to isolate an OOD target test sample from an easily accessible training set, taking convergence speed as an OOD measure. The paper also introduces its decentralized analogue, DIsoN, which counters real-world complications which are encountered in application domains like medical imaging where data privacy and communication constraints rule out accessing OOD detection training sets.

**Questions:**

[Q1] In practical application, communication latency and volatility across both ends may have an impact on the efficiency of DIsoN. Has the paper discussed how such practical network conditions would have an impact on convergence speed and OOD detection efficacy?

[Q2] The statistical significances of OOD detection results are significant for establishing method robustness. Although there is some warrant for computational cost for running multiple seeds on all configurations, would it be possible to report mean and standard deviation for significant experimental results such as comparisons between CC-DIsoN and best baseline between Tables 1 and 2? Reporting them would greatly enhance confidence of findings.

**Ethical Concerns:**

["NO or VERY MINOR ethics concerns only"]

**Final Justification:**

The paper presents practical value, and therefore we have given it a positive score.

**Limitations:**

Yes

**Quality:**

2

**Strengths And Weaknesses:**

Strengths

1. The paper is written in a relatively fluent manner. Its structure is logically organized, and the language used throughout is clear and accessible, making it easy for readers to follow

2. The paper is on OOD detection's practical challenges for the application of medical image analysis, where data transmission and privacy make it hard to maintain direct access to training data at deployment's side.

Weaknesses

[W1] DIsoN only processes one target sample at any given time. While its authors envision future development for handling multiple samples at once, that limits its effectiveness for most batch processing environments.

[W2] DIsoN creates considerable computational overhead at inference time and can range from 40 sec to 4 min on each sample. Such latency can be a significant limitation on applications where fast or real-time inference is needed.

[W3] The manuscript says that α=0.8 performs well on all experiments and validates its effectiveness via ablation experiments. It lacks, though, an indepth theoretical study to guide optimal α selection or an automatic selection mechanism.

---

> ### Author Rebuttal · Authors · 2025-07-31
>
> Dear reviewer, thank you for the constructive and valuable feedback to improve our paper. We are glad you acknowledge the **novelty of the method as original** ("an
> original OOD detection method"), introducing the Isolation Network
> (ISoN), a novel algorithm that trains a model to separate a
> test sample from training data, using convergence to determine
> if the test sample is OOD. We appreciate recognizing the
> value of our introduced decentralised algorithm (DISoN) for
> **addressing real-world practical challenges** ("counters
> real-world complications") of OOD detection in privacy-sensitive settings like healthcare. We are happy you found our paper "**logically structured**" and
> "**easy to follow**".
> The reviewer raised few points about computational
> efficiency, hyperparameter configuration, and confirmation of performance stability across seeds. We address your concerns
> below and have revised our paper accordingly.
>
> ## \[W1\]: Processing one target sample at a time limits batch-processing.
>
> We understand the reviewer's consideration that processing one target
> sample at a time may seem computationally suboptimal by not allowing
> batch-processing. This is not entirely true, however, and computational resources can be optimized by parallelism, if
> required. We want to emphasize that many (if not
> most) applications require inference on 1 sample at a time. For
> example, 1 patient is scanned and the image needs AI processing
> to detect pathologies as soon as possible, without waiting for a batch
> of images from other patients. Similarly for applications like a user requests classification of a bird photo they just took. Thus, we designed our novel OOD algorithm with the requirement to process 1 target sample at
> a time, by learning a decision boundary that
> separates 1 target sample from the training data, using convergence as OOD score.
>
> That said, if a deployment setting allows processing multiple datapoints simultaneously
> (e.g. a pre-collected batch of scans from multiple patients), multiple samples can simply be processed via separate instances of our algorithm in parallel. This isn't a traditional "batch" tensor in Deep Learning libraries, but enables parallel execution of independent processes, making full use of available hardware. In our experiments, the moderate model size allowed up to 8 DIsoN runs in parallel on a single NVIDIA RTX
> A5000 GPU (24 GB VRAM). Our
> code includes scripts to run multiple samples in parallel. We aim to develop more batch-effective variants in Future Work. However, the current work is an original and already effective
> algorithm that makes a novel contribution to knowledge, warranting
> publication to enable further exploration and extensions by the
> community of this type of decentralised OOD detection framework.
>
> NeurIPS allows 1 extra page for post-rebuttal
> additions to the camera-ready, we've added the above discussion to clarify. Thank you for the great point.
>
> ## \[W2\]: Computational overhead at test time, suboptimal for fast or real-time inference
>
> As the reviewer correctly notes, DIsoN introduces some latency per
> target sample at inference (40 sec - 4 mins /sample in our settings).
> Indeed, this novel algorithm is not intended for real-time inference.
> However, a lot of applications don't require
> real-time inference, whilst a few seconds (or even minutes/hours) are
> acceptable to gain enhanced reliability by detecting unexpected OOD
> samples. In many safety critical medical workflows,
> diagnostic accuracy is prioritized over speed.
>
> Current medical workflows involve patient scans waiting hours/days before radiologists review them, due to limited personnel.
> Some scans can wait for a month! If AI is used to
> automate the process or create diagnostic predictions to inform
> humans, integration in the workflow is possible even if
> it needs minutes, or hours/days, as long as it is reliable. The same
> applies to many applications outside healthcare. Thus, we argue the
> method is applicable to a variety of applications, and this should not
> be considered as a reason for rejection of the work.
>
> To help practitioners assess DIsoN's suitability for their use case, we have **added a detailed runtime analysis to the
> Appendix**. The table below summarizes the min/max/median runtime in seconds (s) for ID and OOD samples, plus statistics of communication rounds (R). For brevity, we show only the Ultrasound dataset here. Full results for all datasets are in the updated Appendix.
>
> |Sample Type|Minimum (s)|Median (s)|Max (s)|Mean (R)|25th Percentile (R)|Median (R)| 75th Percentile (R)|
> |-|-|-|-|-|-|-|-|
> |ID|28.2|106.4|274.3|48.2|20.0|29.0|100.0|
> |OOD|18.7|72.8|212.1|28.1|16.0|20.5|32.0|
>
> We also **added a "practical considerations" sub-section** to
> the paper, incorporating these points. Thank you for raising
> this point, we believe the added runtime and
> explanation improves clarity.
>
> ## \[W3\] Theoretical study to find optimal alpha hyperparameter.
>
> $\alpha$ (Eq.2) is indeed a hyperparameter of the algorithm,
> configuring how much the global model should be updated according to the
> updates from the source training database (Server) versus the
> target sample (target node). Like many ML methods hyperparameters, its value needs to be chosen empirically. We don't believe that deriving a theoretical framework for finding an optimal value is possible. In these cases, we believe ideally an algorithm should be relatively robust to choice of values, with similar values performing reasonably well across tasks (i.e., hyperparameter values "generalizing" in some sense). This is why we provided an empirical study for $\alpha$ values (Fig.4.a), showing that across studied tasks, performance remains quite robust
> for a wide range of $\alpha$ values. We see little performance change for
> $\alpha$ values $[0.6-1.0]$, which is a wide range considering
> $\alpha$ is $[0,1]$. Purpose of this sensitivity-analysis
> was not to show that value of $\alpha=0.8$ is optimal, although we
> understand it may be misread like this. Instead, we wanted to
> provide empirical evidence (in absence of theoretical)
> that values in such a range are expected to be reasonable for new,
> unseen tasks, given that they are reasonable for all studied tasks.
>
> We added explanation about the above in the interpretation of Fig
> 4.(a) results, to inform readers about possible reasonable $\alpha$ value choices. Given that many ML methods' hyperparameters are empirically chosen
> (without theoretical framework), and
> our algorithm shows reasonably robustness to the $\alpha$
> value, we believe this does not constitute a strong weakness of the
> algorithm.
>
> ## Questions
>
> ## \[Q1\] How (internet) network volatility may affect the method: This
> is a good practical question. Since the algorithm is designed to
> "serve" only 1 client per process, it's much simpler to manage
> than multi-client decentralised systems (e.g. federated consortia with
> 10s/100s of clients). If the network is slow/unstable, the server
> would simply "wait" for their single client to respond/reconnect. As DIsoN's OOD score depends on successful communication
> rounds for convergence, physical network delays would
> only affect **wall-clock time**,
> which cannot be avoided for services over the internet.
> Fortunately, it would **not compromise convergence quality or
> optimization,** or the number of rounds needed for it. Thus our OOD
> score is robust against this. If the client disconnects, the
> process can continue or restart when it's available
> again (each training session is short, up to 4 mins in our experiments,
> so minimal time lost). Production systems would require timeouts/retry mechanisms, but this is standard practice in
> decentralised systems engineering and beyond the paper's scope, which focuses on
> algorithmic development. We added discussions about these practical considerations to the paper.
>
> ## \[Q2\] Re-run multiple seeds to enhance confidence on performance:
> We absolutely agree that statistical significance is important and
> should be reported whenever possible. We want to sincerely thank the
> reviewer for acknowledging the computational cost (we wish
> every reviewer did this!), which not all labs can afford, thus this
> consideration helps with inclusivity in science. During the limited rebuttal time, we managed to run our CC-DIsoN and the strong SOTA method (PALM) with three seeds each, reporting the mean and standard deviations, as requested. Due to the large
> MIDOG table size, we only provide here the average scores for
> near OOD and far OOD over all tasks (averages of means and stds). We
> updated the full table in the paper:
>
> |Method|Avg. Near AUROC ↑|Avg. Near FPR ↓|Avg. Far AUROC ↑|Avg. Far FPR ↓|
> |-|-|-|-|-|
> |PALM|$61.3\pm4.4$|$88.8\pm2.9$|$98.3\pm2.3$|$11.6\pm19.1$|
> |CC‑DIsoN|$69.6\pm1.7$|$81.0\pm3.3$|$98.3\pm0.1$|$4.4\pm1.6$|
>
> Multiple seed runs for X-Ray, Dermatology, and Ultrasound:
>
> |Method|X-Ray FPR ↓|X-Ray AUROC ↑|Derma. FPR ↓|Derma. AUROC ↑|Ultra. FPR ↓|Ultra. AUROC ↑|
> |-|-|-|-|-|-|-|
> |PALM|$85.7\pm9.2$|$65.4\pm6.5$|$62.4\pm0.8$|$77.3\pm1.0$|$75.6\pm5.5$|$59.4\pm3.1$|
> |CC-DISoN|$61.9\pm1.2$|$84.9\pm0.9$|$42.5\pm4.4$|$89.5\pm1.5$|$73.2\pm4.7$|$65.6\pm1.2$|
>
> We will run multiple seeds for all methods until camera-ready and update Tab2 and 3. Our method's average performance is higher than
> the best baseline by a considerable margin and a small standard deviation over 3 seeds, providing
> further confidence in our findings. Though we note it should be interpreted with
> caution given it is calculated with only 3 seeds.
>
> We thank the reviewer again for the constructive feedback! It
> helped us improve the paper and clarify these considerations. We
> hope this allows you and readers to better appreciate the strengths
> of our novel framework for decentralized OOD detection services, deviating from current approaches, that we believe should
> be communicated to the community to inspire further developments. Please
> let us know if any concerns remain. We are happy to engage in
> further discussion!

---

> > ### Comment · Reviewer_SHNi · 2025-08-04
> >
> > Thank you for your response — it has helped clarify many of our concerns. One remaining question we have relates to computational cost: have you compared the efficiency of your method with relevant baselines? Since timely results can be critical in healthcare scenarios, this would be a valuable aspect to report. Nevertheless, we find your work to be of high value, and we will maintain our positive score.

---

> ### Author Response · Authors · 2025-08-06
> **Clarification on the remaining question**
>
> Thank you for the follow-up! We are glad to hear that we have addressed
> most of your concerns and that you **"find \[the\] work to be of high
> value"**. We believe the work has an original idea with intriguing
> properties, so since you seem to have found it valuable and most
> concerns addressed, we really hope to receive your stronger support with
> an increased score!
>
> We're happy to provide more details about this remaining question on
> computational cost. We provided what we thought is most useful in our
> previous response \[W2\] but due to space limits we may missed
> something. Glad you asked for further details!
>
> Follow-up question:
>
> > *"have you compared the efficiency of your method with relevant
> baselines?"*
>
> We measured inference times and now included results in the Appendix for
> all methods and databases. Below are **inference times per sample** for
> the most relevant methods for the Ultrasound data. For DIsoN we report
> separately for ID and OOD, which differ because our OOD score depends on
> time to convergence. These times are hardware dependent (NVIDIA A5000
> here) but allow comparison and inform readers.
>
> |Method|Avg. (seconds)|
> |-|-|
> |MDS|0.006|
> |ViM|0.005|
> |iForest|0.015|
> |Cider|0.007|
> |Palm|0.007|
> |CC-Dison (ID samples)|155.2|
> |CC-Dison (OOD samples)|90.9|
>
> All compared methods use a standard Conv Neural Net (CNN) as backbone
> (ResNet in our experiments). They all first perform one CNN forward
> pass, which only takes milliseconds on moderns GPUs. Then they just
> apply different computationally-light followup steps to compute an OOD
> score: distance in embedding space (MDS, PALM, CIDER), linear projection
> (ViM), or tree traversal (iForest), and therefore retain real-time
> inference. Note that iForest, which is conceptually closest to DIsoN,
> cannot be trained with the target and source training samples jointly
> due to privacy constrains, unlike DIsoN. Therefore, as in the original
> iForest paper \[REF2\], we pre-train it on source data (ID) and then
> apply it with a forward pass on each target sample. This makes it faster
> than DIsoN.
>
> While DIsoN outperforms in terms of accuracy, it requires more inference
> time due to training a CNN per target sample - here up to 2.5 minutes,
> though it can be longer on other databases, as we reported in the original
> paper, and we now added more info in the Appendix.
>
> Follow-up question:
>
> > *"Since timely results can be critical in healthcare scenarios, this
> would be a valuable aspect to report."*
>
> We agree inference time is useful to report to inform readers about
> whether the method is appropriate for their use-case. But allow us to
> clarify the following about healthcare workflows, as it clarifies that
> **our method has high practical significance regardless** the increased
> inference time.
>
> As discussed in our previous response \[W2\], most medical workflows do
> not require real-time inference. Globally, due to heavy workload of
> radiologists, most radiology departments have a \"turn around time"
> (TAT) for scans of hours, days or even weeks between scan acquisition
> and when they are read by a radiologist \[REF1\], except for
> emergencies. If AI would be integrated in such workflows, e.g. to
> provide info to radiologist (here predicted class) or a diagnosis before
> the radiologist, the AI would process scans during this TAT waiting
> time. Hence, in many workflows, the algorithm has between hours or weeks
> for processing, and results would be ready when the radiologist reviews
> the scan. Thus our method is suitable for all these workflows. Faster
> diagnosis is mainly needed in emergencies (minutes to hours) and
> ultrasound (reported real-time). Our method could be used to some of the
> former (if inference time is short enough) but not the latter.
>
> Overall, one of the main factors hindering integration of AI in
> healthcare is limited reliability, rather than inference time.
> Therefore, methods like the proposed that enhance reliability are of
> great significance to the field (and other safety-critical
> applications). Hence DIsoN was designed with performance as main goal,
> not speed. While not suitable for all scenarios (no method is!), it is
> appropriate for many applications as above. Although we don't focus on
> extensive inference time analysis, as it's not our focus, we absolutely
> agree that providing sufficient information about inference time is
> useful for readers to judge if the method is appropriate for their
> applications. Therefore, we **commit to adding, in the updated camera
> ready version of the manuscript,** information about the method's
> runtime, including those provided in our original response \[W2\] and
> this one.
>
> \[REF1\] NHS England, Diagnostic imaging reporting turnaround times,
> 2023 (online).
>
> (NeurIPS guidelines don't allow links)
>
> See table for turn-around times of different scans.
>
> Example for NHS in UK, but other countries have similar delays due to
> radiology over-burden.
>
> \[REF2\] Liu et al., Isolation Forest, 2008

---

### Note · Authors · 2025-08-14

We thank the AC for their time and the reviewers for their constructive feedback that strengthened our work.

We have addressed all main concerns and misunderstandings in the rebuttal. The main points were:
- questions on runtime and practical use-cases
- performance stability across seeds
- misunderstanding that data were synthetic
- comparison to Few-Shot Federated Learning
- questions on class-conditional sampling
- other clarifications, e.g. hyper-parameters

We presented the following experiments and clarifications in the rebuttal and **commit to adding related content to the manuscript**:

- **Runtime:** Added runtime analysis with convergence-round distributions and per-method runtime comparisons to help practitioners assess suitability for their use-case. Clarified DIsoN is suited for intended safety-critical workflows in medical imaging that don't require real-time inference (hours-weeks turnaround where reliability outweighs speed)

- **Performance stability**: Added multi-seed experiments, reporting mean and std. Results confirm stable improvements
- **Misunderstanding on real-world data:** Clarified all data are real-world with naturally occurring OOD artifacts
- **Few-shot FL comparison**: Explained fundamental differences and why direct comparison to our work isn't applicable
- **Class-Conditional sampling**: Added experiments with misclassification scenarios, extending analysis
- **Further improvements**: Refined mathematical formulation and writing, added discussion on hyperparameter choice and fine-tuning

All reviewers agreed their concerns have been addressed after the rebuttal. Reviewers **vx6Z** and **8S2S** stated they raised their scores.

Reviewer **SHNi** stated they "**find (our) work to be of high value**" and will "**maintain [their] positive score**" after the rebuttal (before considering our follow-up response, 6 Aug, to their question during the discussion, 4 Aug). We believe we addressed all points and clarified in our follow-up that our method's inference-time isn't a major weakness as it's well-suited for medical workflows, hence we hope this is reflected in an increased score.

Our work introduces a novel decentralized OOD detection framework for privacy-sensitive domains like healthcare, with reviewers appreciating its originality (rated "Good"-SHNi,8S2S; stated "novel"-8S2S) and significance ("high value"-SHNi). We believe acceptance would be valuable for the community to learn and further study this original framework.

---

### Decision · Program_Chairs · 2025-09-17

**Decision:**

Accept (poster)

**Comment:**

All the reviewers have actively engaged with the discussion and unanimously voted for acceptance. After checking the review, the rebuttal, the manuscript, and the discussion, the AC agrees with this assessment.